# Wide-swath altimetry maps bank shapes and storage changes in global rivers

A. Cerbelaud[1,5 ✉], J. Wade[1,5], C. H. David[1 ✉], M. Durand[2], R. P. M. Frasson[1], T. Pavelsky[3] & H. Oubanas[4]

Rivers are Earth's most renewable and accessible freshwater resource[1], yet global estimates of the magnitude and variability in river water storage have remained few and inconsistent[1–9]. Previous estimates of variability have relied either on sparse and asynchronous remote-sensing observations[10] or on hydrological models constrained by incomplete understanding of surface-water balance and poorly known river channel characteristics[2,3]. The insufficient knowledge of temporal variations in river water storage across space hinders effective management of this critical freshwater resource[11,12]. Here we present near-global-scale observations of active river channel geometry and associated monthly changes in water storage at the reach scale derived from the first water year (October 2023 to September 2024) of the Surface Water and Ocean Topography (SWOT) mission at 126,674 reaches worldwide. Clear patterns of riverbed shape and storage variability expectedly emerge across major basins. SWOT reveals a range of 313.1 ± 129.5 km³ in global annual river storage variability, approximately 28% lower than the lowest previously modelled estimates for the same wide reaches. Although the Amazon's 2024 record drought, the observational challenges in the Arctic and the revisit frequency of SWOT almost certainly contribute to the discrepancy, the observations point to distinct knowledge limitations in surface-water science. These findings highlight key opportunities to improve the fundamental representation of surface-water dynamics in global models and to better inform water resource management and disaster mitigation at scale.

Precise monitoring of global river fluxes and storage is becoming increasingly vital as river corridors and their biodiversity are threatened by global environmental change[13–15], intensifying extreme events[16–19], and rising anthropogenic pressures such as pollution, population growth and transboundary conflicts[11,12,20,21]. Rivers hold only 1% (approximately $2 \times 10^3$ km³) of all liquid surface freshwater by volume (approximately $194 \times 10^3$ km³), which itself accounts for less than 1% of all liquid freshwater on Earth (approximately $23,594 \times 10^3$ km³)[1], but offer the most renewable and accessible freshwater supply, making them an essential component of sustainable water availability for ecosystems and people. Still, few global estimates of global river storage magnitude and variability exist[1–9]. These estimates show limited reliability and a large spread, primarily owing to the challenge of accurately modelling the global hydrological cycle and the lack of observational constraints on global rivers.

Two numerical modelling studies have recently estimated the intra- and interannual variability in Earth's river storage using outputs from global hydrologic models[2,3] and their findings differ by an order of magnitude. This divergence stems from two major uncertainties: land run-off inputs into rivers and water propagation speed, an emergent property of channel geometry[2] and residence time[3]. Run-off and propagation speed are at the core of even the simplest mathematical

models for river water propagation[22,23], yet neither is well constrained at global scales, leading to widespread disagreement in estimates of modelled river water storage and its variability. Alternatively, remote sensing has offered a promising counterpart to modelling for estimating river water storage changes. Common approaches combine satellite measurements of water surface elevation (WSE) from altimeters with river width from optical or radar imagers, and sometimes incorporate digital elevation models, producing empirical elevation–width relationships at individual river cross-sections[10]. These satellite-based storage change estimates have suggested very limited agreement with hydrologic model outputs, underscoring that large uncertainties are likely to affect global estimates[10]. One key limitation of past remote-sensing approaches is that elevation measurements have been opportunistic and sparse, following the widely spaced ground tracks of ocean-tailored altimetry missions. Another limitation is that elevation and width have typically been observed asynchronously from different satellite platforms, hence requiring temporal interpolation and introducing additional observational uncertainty[24,25].

Launched in December 2022 after 20 years of development[26–29], the Surface Water and Ocean Topography (SWOT) satellite is a space mission specifically designed to observe Earth's continental surface waters

[1]Jet Propulsion Laboratory, California Institute of Technology, Pasadena, CA, USA. [2]School of Earth Sciences, The Ohio State University, Columbus, OH, USA. [3]Department of Earth, Marine and Environmental Sciences, University of North Carolina, Chapel Hill, NC, USA. [4]G-EAU, Univ Montpellier, AgroParisTech, BRGM, CIRAD, INRAE, Institut Agro, IRD, Montpellier, France. [5]These authors contributed equally: A. Cerbelaud, J. Wade. ✉e-mail: arnaud.cerbelaud@jpl.nasa.gov; cedric.david@jpl.nasa.gov

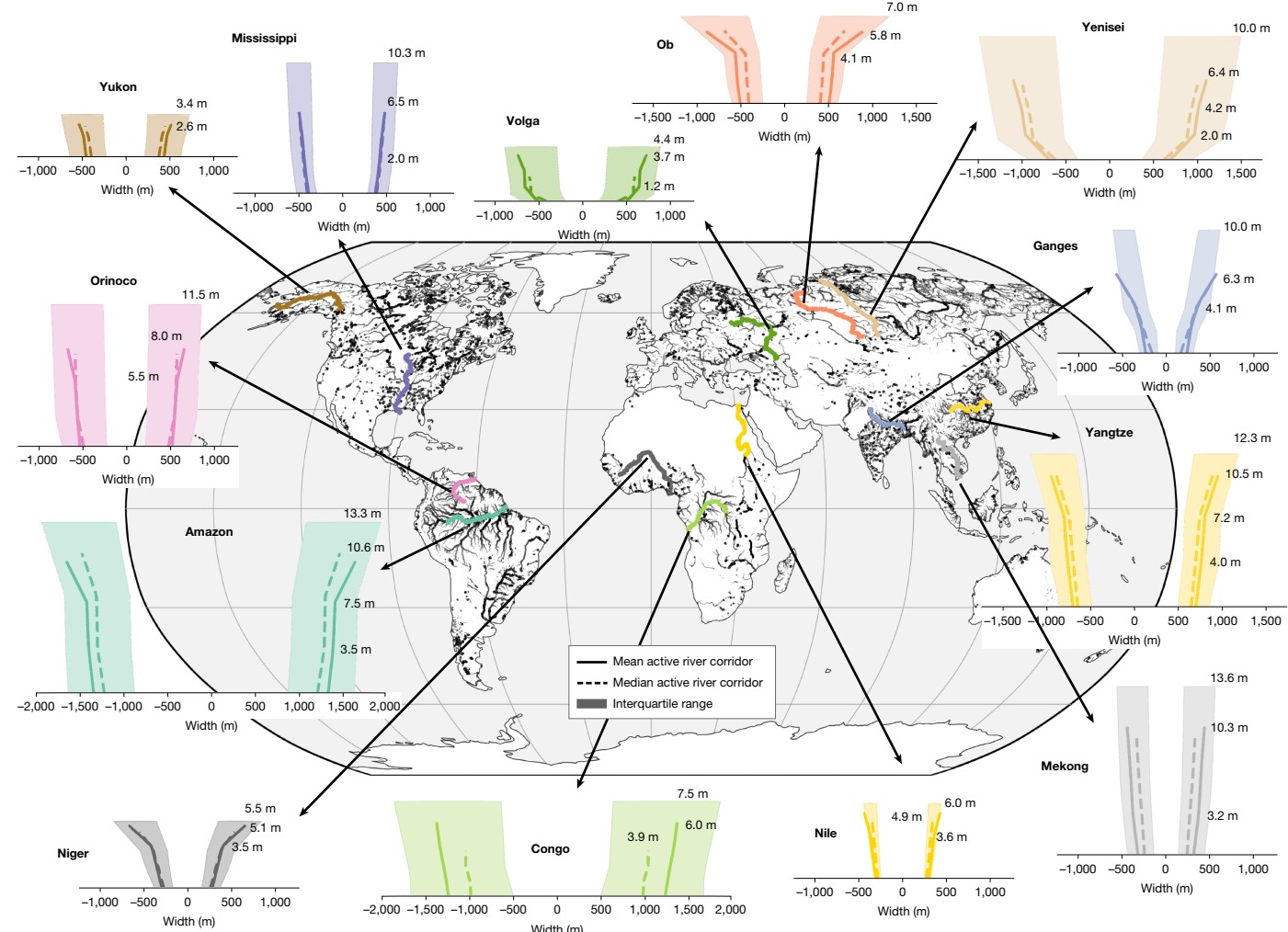

**Fig. 1 | SWOT observes the shape of active river corridors along major global rivers.** The shapes result from 1 year of SWOT measurements between the lowest and highest observed water levels. To illustrate the river corridors, the measured widths are given along the *x* axis centred around zero, and heights are given along the right bank side. The mean, median and interquartile range represent the reach-level spatial variability in active river corridor shapes along the river. Basemap from Natural Earth (https://www.naturalearthdata.com).

including rivers, lakes, reservoirs and wetlands[30]. SWOT has already demonstrated its ability to capture finer-scale ocean dynamics[31] than had been observed by three decades of sea-surface-height observations from satellites. To do so, SWOT uses its Ka-band radar interferometer[32,33] to simultaneously map water extent and associated WSE over Earth's widest continental water bodies. SWOT observations of water extent and associated WSE in rivers are gathered on a hydrographic network[34] of approximately 200-m-spaced nodes and approximately 10-km-long reach segments for rivers wider than 30 m. These concurrent observations can be used to map the active beds of river corridors and derive global river water storage variability, a capability long anticipated[35] but never previously demonstrated globally, thereby shedding light on the first hydrologic science question that SWOT was designed to help answer[30,36]: "What are the temporal and spatial scales of the hydrologic processes controlling surface-water storage and transport across the world's continents?".

Here we present near-global measurements of the shape of active river corridors derived from joint SWOT observations of river width and elevation from the first water year of SWOT's science orbit (October 2023 to September 2024). We also provide a near-global observational record of river storage variability. The observations reveal that rivers with comparable discharge can exhibit markedly different channel shapes and highlight the wide morphological diversity

of river systems across the globe. The observed storage changes help identify and quantify clear annual cycles and hotspots of global river storage variability. Although marked by early observational and technological uncertainties, this record of storage change, when contrasted with recent model results[3], emphasizes the fundamental improvements required to represent surface-water dynamics in global models with implications for water resources management and water-related disaster mitigation alike.

## Shape of active river corridors

Using about 1.65 million carefully filtered SWOT observations from the mission's first water year, we derive relationships between river width and WSE for 126,674 river reaches, covering 73% of the world's widest rivers. By fitting reach-scale piecewise linear functions[37] that account for measurement errors[38], we map how river width changes with water levels, providing near-global characterization of river corridor shapes between the lowest and highest observed water levels, that is, the 'active' shape, from satellite observations (Fig. 1). The diversity in active riverbed shapes observed by SWOT—ranging from concave to convex[39], steep to gentle, and stable to highly variable cross-sections—is both expected and striking. These findings reinforce long-standing concerns about the limitations of using uniform channel

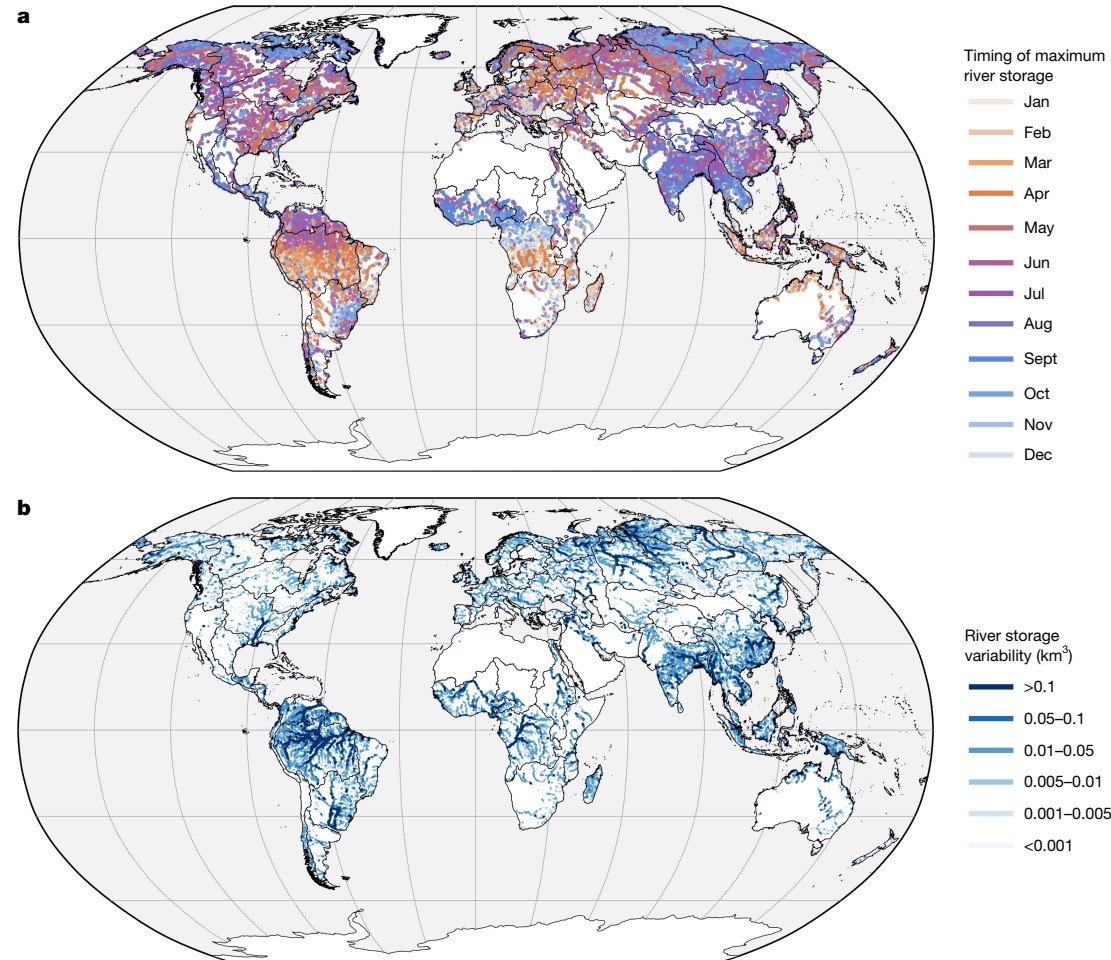

**Fig. 2 | SWOT captures river storage variability near-globally between October 2023 and September 2024. a**, SWOT-observed timing of the peak RSA. **b**, SWOT-observed annual river storage variability (ΔRSA). Basemap from Natural Earth (https://www.naturalearthdata.com).

geometries such as trapezoidal or rectangular forms in global river models. The Amazon, Orinoco, Yangtze, Ganges, Mekong, Mississippi and Yenisei rivers all exceed 10 m in observed peak-to-trough water level variability.

Located in contrasting climates, the Mississippi River showcases much smaller spatial variability than the Yenisei River despite comparable average discharge and both being highly managed. The Orinoco River is much narrower and steeper than the Congo River although they carry similar amounts of water to the ocean. Varying interquartile ranges indicate contrasting geomorphologies along river courses (for example, Amazon versus Congo basins). These results offer a promising glimpse into the diversity of active river corridor shapes across Earth's widest rivers. However, the shapes only reflect what SWOT observed during 2023–2024. Notably, floodplains appear to be rarely captured in the elevation–width relationships (Fig. 1). The results also reflect early-stage limitations in both our three-domain piecewise approach and SWOT's current measurement capabilities and uncertainties. These measurement characteristics are currently being investigated in the context of large-scale in situ validation efforts[40]. In addition, 1 year of observations at an average repeat of 28 days (after data filtering; Extended Data Fig. 1) probably obscures occasional high extremes, and the absence of data over frozen water induces biases in elevation–width relationships for Arctic rivers. As the mission progresses, uncertainties decrease, analysis methods evolve and finer-scale products are leveraged, we anticipate notable improvements in resolving channel geometry at both global and local scales.

## Global river storage change from SWOT

From the integrated width–elevation relationships, we derive reach-scale cross-sectional area changes, propagate measurement and regression uncertainties[10], and convert these to volume changes relative to a reference WSE. After interpolating over SWOT overpass dates that were filtered out, we compute zero-mean monthly river storage anomalies (RSA) at reach, basin and global scales, characterizing storage changes observed by the satellite. However, the full river water storage remains unresolved.

Our time series of reach-specific RSA derived from the initial water year of SWOT at 126,674 reaches represents a near-global observational dataset of monthly variability in river storage, with uncertainty. The approximate interpolated 10-day revisit rate and fine spatial coverage offered by SWOT allows us to study the spatial patterns of time-varying estimates of RSA. We visualize the seasonal timing of storage variability regionally by plotting the month when each reach experiences its highest RSA within the annual cycle (Fig. 2a). Although RSA timing is rather spatially noisy, seasonal patterns generally follow latitudinal hydroclimatic gradients. In the Amazon River Basin, river volume tends to peak in March to May, trending towards June northwards of the basin (Fig. 2a), consistent with previous understanding[41]. SWOT-observed peak river storage occurrence for the Congo River Basin also coincides with previous knowledge[42], occurring in October–December in the northern and central parts of the basin, and around March–April in the south.

To visualize spatial patterns of the amplitude in global river storage variability, we quantify the annual range of RSA (ΔRSA, along with

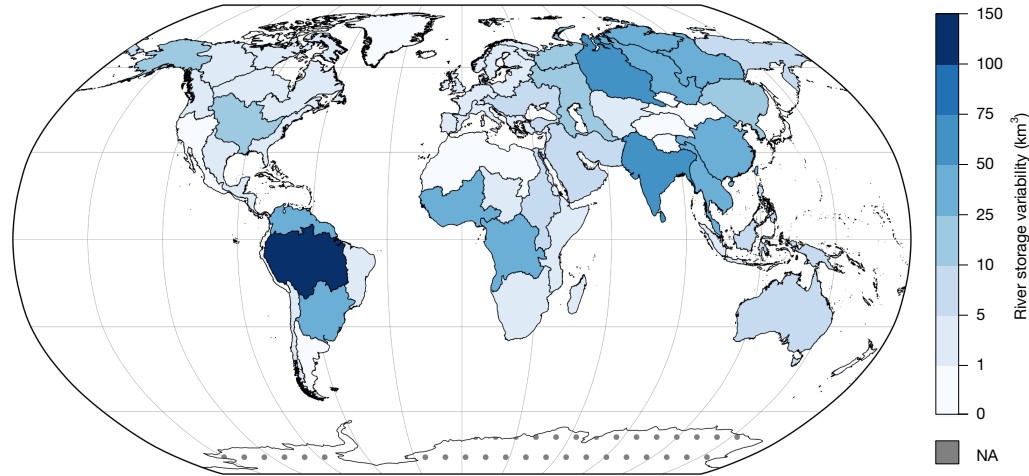

**Fig. 3 | SWOT provides near-global basin-scale river storage variability between October 2023 and September 2024.** River storage variability (ΔRSA) is defined as the annual range of monthly volume anomalies in each of 61 basins in the SWOT river database. NA, excludes Antarctica. Basemap from Natural Earth (https://www.naturalearthdata.com).

its uncertainty) by computing the difference between its maximum and minimum monthly values. At the reach scale, SWOT observes expected regional hotspots of global ΔRSA in the world's largest river systems given close links to upstream area, river size and precipitation regimes (Fig. 2b). The global spatial mean and median of ΔRSA magnitude in SWOT-observed reaches are $0.013 ± 0.0012$ km³ and $0.003 ± 0.0005$ km³, respectively. The largest rivers by flow (for example, the Amazon, Ganges–Brahmaputra, Congo, Yangtze, Mississippi) show the greatest variability as anticipated, with variability increasing downstream within individual basins. These expected patterns help lend some confidence to both the early SWOT observations and the processing methodology applied here.

At the basin scale (Fig. 3), we find that the Amazon River Basin shows a ΔRSA of $172.9 ± 16.4$ km³, the world's largest by-basin value, consistent with past studies of global river storage variability[2,3]. Following the Amazon, the basins with the largest aggregate ΔRSA are the Ob ($55.0 ± 8.1$ km³), the Indian subcontinent ($52.0 ± 8.$ km³), the Orinoco ($46.5 ± 5.0$ km³) and the Congo ($37.3 ± 4.8$ km³), and are further illustrated in Extended Data Fig. 6. Strikingly, the SWOT observations of the Nile River Basin yield a ΔRSA of only $8.5 ± 1.6$ km³ in 2023–2024, much lower than previous estimates (Extended Data Figs. 6 and 7). This discrepancy may stem from limitations in previous estimates, severe drought in the upper basin—particularly the Blue Nile—during the 2023 peak season (late summer), and possibly from water retention by the Grand Ethiopian Renaissance Dam (active capacity of about 60 km³), which could reduce overall ΔRSA. However, a recent study did not reach definitive conclusions on the downstream effects of the dam[43].

Despite the complexity of validating such unprecedented fine-scale global measurements of width and WSE (and therefore of RSA), we quantify SWOT's RSA consistency using 61 global in situ monthly discharge gauges, representative of each SWOT global basin. Seasonal validation confirms that SWOT RSA appears reliable across most of the globe, except for parts of the Arctic, southern South America and the western USA (Extended Data Fig. 3). Low correlation with in situ discharge in the Arctic is aligned with the lack of usable observations over frozen rivers and the occasional (probably erroneous) observation of high storage in the autumn outside the spring freshet period (for example, in Eastern Siberia; Fig. 2a). Overall, the limited effective temporal sampling of SWOT can cause the monthly and annual RSA ranges to miss part of the true variability.

In addition to in situ validation, we conduct a global statistical analysis of the SWOT-derived ΔRSA using a tailored outlier detection approach. Our findings highlight reduced reliability of RSA estimates at

3.4% of global river reaches, accounting for 8.8% of ΔRSA but impacting average seasonality only by a 1.7-day shift. These outliers are mostly located in complex and dynamic riverine environments (for example, braided systems and deltas), consistent with known limitations in the SWOT hydrographic network (such as centreline positioning quality; Supplementary Information section 5). Continued SWOT observations will refine understanding of its observational perimeter, processing algorithms and uncertainties, ultimately helping to better constrain RSA accuracy. While assessing compliance with SWOT mission requirements is beyond our scope, we hope that these results will contribute to advancing the mission's scientific objectives.

## Comparison with previous simulations

The sum of RSA values for all 126,674 reaches in this study provides a near-global estimate of river water storage anomalies (Fig. 4a). The near-global standard deviation of RSA is 107.8 km³ and its peak-to-trough amplitude (that is, range) is ΔRSA = $313.1 ± 129.5$ km³. The world's rivers wider than 50 m are estimated to account for about 96% of global river storage variability[44]. The SWOT hydrographic network was built from global rivers wider than 30 m, and this study accounts for 73% of these rivers. We therefore anticipate our analysis to capture a substantial portion of global river storage variability (Supplementary Information section 2).

Until now, estimates of this global quantity relied entirely on model simulations, with only two studies hereafter reporting global values. Global simulations including floodplains[2] estimated surface-water storage variability of 3,680 km³. Modelling without floodplains but correcting for known bias estimated the standard deviation of RSAs as ranging from 288 km³ to 721 km³ depending on 3 scenarios for flow wave celerity[3]. SWOT observations of river water storage variability therefore appear much lower than had previously been anticipated. However, the difference between SWOT observations and model estimates are best analysed over a comparable perimeter and set of river reaches.

We leverage a translational dataset[45] to identify modelled river reaches that directly correspond to the SWOT observations, together with publicly available river water storage (without floodplain) model estimates[3], to provide context from previous knowledge (Fig. 4a). Even at corresponding reaches, our SWOT-based global ΔRSA estimate ($313.1 ± 129.5$ km³) is consistently smaller than the 30-year mean (1980–2009) model estimates of all 3 scenarios[3], including 28.2% less than the lowest-volume scenario (ΔRSA of 436.7 km³), thus challenging previous expectations. However, SWOT-observed global monthly time series

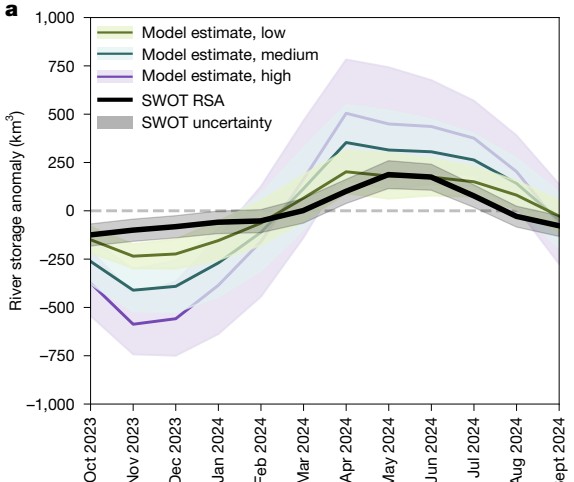
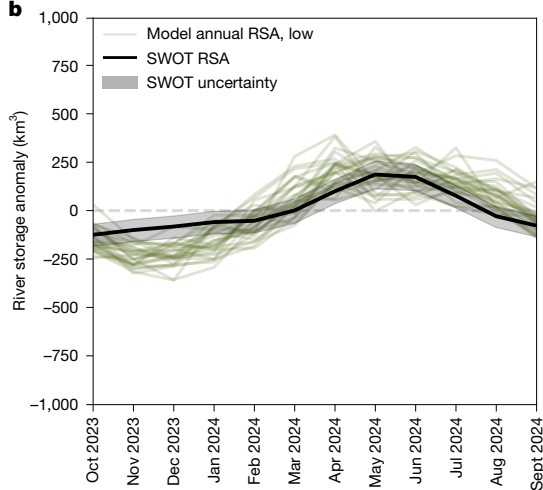

**Fig. 4 | Near-global comparison of monthly river water storage changes from SWOT and previous knowledge.** The shaded regions represent the RSA uncertainty envelopes. Previous knowledge is derived from model simulations described in ref. 3. Modelled RSAs are aggregated at SWOT corresponding reaches and cover the years 1980 to 2009 (inclusive). **a**, Comparison with the 30-year record of 3 residence time scenarios, with the central line representing the 30-year monthly mean bounded by an envelope defined by the monthly standard deviation of simulations. **b**, Comparison with the 30 annual RSA slices of the low-volume-scenario simulations. Excludes Antarctica.

of RSA are consistent with the 2 least variable years of the low-volume simulations (1984–1985 and 1985–1986 with $\Delta$RSA $\lesssim$ 400 km³; Fig. 4b) and generally follow the anticipated annual cycle (Fig. 4 and Extended Data Figs. 9 and 10). Temporal correlation and underestimated magnitude against the previous low-volume scenario[3] are both also apparent regionally, and inconsistent patterns and/or ranges were found in several Arctic basins, Central Asia, southern South America, the Nile Basin and the western USA (Extended Data Figs. 6, 7, 9 and 10). The regional contribution to monthly offset between SWOT and modelled RSA is visualized in Extended Data Fig. 6. Exceptional drought conditions in the Amazon Basin since 2023[46] very likely impacted the observed global $\Delta$RSA (Extended Data Fig. 6) as several major rivers fell below historical records[16,47], and conditions continued to worsen in 2024[48]. In addition, the Lena and Yenisei basins account for most of the discrepancies between SWOT and the model from November to March (Extended Data Fig. 6). SWOT-based variability for 2023–2024 is probably rationally lower than that of past climatological averages. In fact, the world's river conditions in 2024 were reported 'much below' average in South Africa, the Congo, the Nile, the Mackenzie, southwestern North America, the Amazon and the Lena, in strong agreement with SWOT's underestimations against the prior model (Extended Data Fig. 7 versus Fig. 7 of the 2024 World Meteorological Organization *State of Global Water Resources Report*[49]). A longer data record from SWOT's planned 3-year duration will help refine the elevation–width approach and measurement uncertainties, and allow the analysis of potentially more representative hydrological years. The inconsistency observed between SWOT-derived and modelled RSA may also originate, in part, from contrasted definitions of river storage between SWOT's observational perimeter and the model structure (Supplementary Information section 3).

The discrepancies also expose a deeper issue: global river models rely on poorly constrained estimates of land run-off and flow wave celerity; parameters that are difficult to measure and often entangled during model calibration. As a result, simulated river volumes may reflect compensating errors rather than physical reality. For example, overly low celerity values may be used to attenuate run-off peaks, hence increasing residence time and inflating storage. Spatiotemporally uniform celerity values may also be overly simplistic, as they have been shown to vary widely with topography and flow conditions[50]. SWOT-based river storage change estimates still face numerous uncertainties, but the mission offers a path forward: with more observations across annual cycles, uncertainties may be untangled in the hope to build more physically grounded, globally consistent models of river dynamics.

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

## Methods

### SWOT river observations

The SWOT mission uses a wide-swath Ka-band radar interferometer (KaRIn) to collect global measurements of key terrestrial surface-water features, such as extent, elevation and slope, with an average revisit frequency of 10 days over its full 21-day cycle[27,30,32,33]. In this study, we use observations from the SWOT Level 2 KaRIn High Rate River Single Pass data product (RiverSP, version C PIC0 and PGC0), which provides node-averaged and reach-averaged measurements of river elevation and width from individual satellite passes[51]. We use the reach-averaged measurements, with a nominal length of 10 km. The RiverSP observations are aggregated to vector reaches in the SWOT River Database (SWORD)[34] v16, which contains approximately 240,000 global river reaches wider than about 30 m.

We retrieve single-pass SWOT observations of river width, WSE and associated observation attributes at all global SWORD reaches from 1 October 2023 to 30 September 2024, corresponding to the satellite nominal orbit with a 21-day cycle, using NASA PO.DAAC's Hydrocron API service. To ensure the reliability of our analysis of SWOT observations, we apply a series of data quality filters to remove potentially erroneous measurements. We only compute river storage change at SWORD reaches representing rivers (type 1 and type 5), removing reaches corresponding to lakes, reservoirs, dams and waterfalls, and ghost reaches[34]. We acknowledge that as SWORD reflects reservoir extents at the time of its input compilation, reservoirs constructed more recently may be treated as river reaches, potentially introducing local uncertainties in SWOT data. We filter out SWOT observations with a summary quality indicator (reach_q) of 'bad' (3% of observations), retaining observations deemed 'good' (0.03%), 'suspect' (31%) and 'degraded' (66%). We only retain SWOT observations with 'good' cross-over calibration (xovr_cal_q = 0). We exclude those affected by dark water conditions (dark_frac > 0.3), and reach-level measurements with substantial missing node data (obs_frac_n < 0.5). Observations where external climate indicators imply the presence of ice or snow (ice_clim_f > 0) are also filtered. As SWOT struggles to produce accurate observations both in close proximity and far from the nadir track of the satellite owing to the angle of radar echoes, we filter observations with a cross-track distance within 10 km or outside of 60 km of the nadir track. In some cases, the range of WSEs measured by SWOT at a given reach are implausibly large. We therefore filter SWORD reaches from further storage computation if the range of elevation observations exceed 20 m, which surpass most of the largest flood height variations observed in the Amazon[52]. After filtering, we only compute river storage at the 126,674 SWORD reaches with at least 5 valid SWOT observations of WSE and width during the study period (Extended Data Table 1).

We note that the SWOT version D products, forward-processed from April 2025 onwards, resolve several known issues in the version C products used in this study. However, the reprocessing of version D products for science orbit observations between July 2023 and April 2025 is only scheduled to complete in 2026, so that only one full annual cycle is available at the time of this study (autumn 2023 to autumn 2024).

### Effective SWOT sampling after quality filtering

The filtering applied to the initial SWOT dataset results in a reduced effective sampling rate, which varies depending on the configuration and quality of the observations. To assess the temporal coverage of the filtered dataset, we evaluate the effective measurement timeline at the basin level (Extended Data Fig. 1). On average, after quality filtering, SWOT provides 1 valid observation every 28 ± 4.6 days (mean ± standard deviation across basins), with well over half of the expected measurements being excluded during the filtering process. A key limitation arises in regions affected by seasonal ice cover, such as over Arctic rivers and high mountainous basins, where no acquisitions are retained from approximately November to May. This absence of data in winter results

in annual time series that are biased towards summer and early autumn months, and are likely partly unreliable or misleading for hydrological analysis.

### River hypsometry

The simultaneous observations of river width and WSE by SWOT offer a valuable opportunity to map the complex active bathymetry of river channels across a range of flow conditions. River hypsometric curves capture the typically monotonic increase between width and WSE and, importantly, enable the calculation of channel cross-sectional area through integration[37]. By fitting these hypsometric curves to SWOT observations of width and WSE at individual SWORD reaches, we can track changes in channel flow area over time and convert these area estimates into estimates of storage variability. The cross-sectional area of a river channel as observed by SWOT can be represented by the function:

$$A(H) = \mathring{A} + \int_{H_{\min}}^{H} W(H')\mathrm{d}H' \tag{1}$$

where $H$ is the WSE, $W$ is the river width, $H_{\min}$ is the lowest WSE observed by SWOT, $\mathring{A}$ is the unobserved cross-sectional area below the lowest observed WSE, and $W(H')$ is the hypsometric function between width and WSE[37,53,54]. SWOT is unable to capture the entire wetted channel below baseflow levels. Therefore, a portion of the channel cross-section, denoted as area $\mathring{A}$ in Extended Data Fig. 2a, remains unobserved below the lowest recorded river height. Although the official SWOT mission discharge product currently involves the estimation of full channel geometry[53], these estimates require further validation at the global scale. Therefore, we do not attempt to use them here. We note that equation (1) is only an approximation as measurements are averaged at the reach scale, overlooking finer-scale longitudinal variations in width and height. This approach forms the basis for the estimation of river discharge from SWOT[53] and has been implemented in the Confluence software engine (https://github.com/swot-confluence/) used to produce SWOT discharge products and the Flow Law Parameter Estimation library FLaPE-Byrd (https://github.com/mikedurand/FLaPE-Byrd).

As SWOT observations of both width and WSE include associated measurement error, standard regression approaches are not suitable for accurately fitting hypsometric functions. Instead, we fit river hypsometric curves to SWOT observations of width and WSE using an 'errors-in-variable' approach, as implemented by the FLaPE-Byrd repository and detailed in ref. 37. Originally introduced in ref. 38, errors-in-variables alters the standard least squares regression to explicitly account for measurement uncertainty in both width and WSE[37]. Reference 37 showed that constraining elevation–width observations with hypsometric curves using errors-in-variable regression reduces the variance in both measurements and improves precision of cross-area estimates. Although measurement uncertainty is included in the SWOT product for each river observation, we find that at the time of this study, the mission's error estimates remain unreliable. When applying the errors-in-variable approach, we use an assumed WSE uncertainty of 0.1 m and a width uncertainty of 30 m, reflecting the science requirements of the SWOT mission[30]. A sensitivity analysis of active riverbed shapes and resulting RSAs to SWOT width and elevation uncertainties is presented in Supplementary Information section 4, documenting the impact of these input uncertainties.

Owing to the evolution of channel bathymetry across flow regimes, the relationship between river width and WSE is frequently nonlinear[37]. Following ref. 37, we fit a three-part piecewise linear relationship to SWOT observations of width and WSE of the form:

$$W(H) = \begin{cases} y_1 + m_1H & H < H_1 \\ y_2 + m_2H & H_1 \leq H < H_2 \\ y_3 + m_3H & H_2 < H \end{cases} \tag{2}$$

where $y_i$ are the intercepts of the linear regression segments, $m_i$ are the slopes of the linear regression segments and $H_i$ are the WSE breakpoints between subdomains for regions $i = 1, 2, 3$ (Extended Data Fig. 2b). The subdomains ideally reflect distinct hypsometric relationships across three potential flow regimes: within-bank flow, the transition to out of bank flow and out of bank flow (floodplain)[37]. Depending on what SWOT sampled, only within-bank flow may have been captured or kept post-filtering, and the three subdomains will then represent different in-bank shapes. When fitting the hypsometric curves, the piecewise linear segments are constrained to be continuous. In some circumstances, the initial errors-in-variable regression fails to converge or produces implausible results ($m_i$ terms either greater than 10,000 or negative), often owing to unreliable width measurements. In such situations, we instead fit a simplified rectangular hypsometric curve using the reach's median width. This approach relies on the more reliable WSE observations to capture flow variability, although it sacrifices detail in representing the channel's bathymetry. Of the 126,674 SWORD reaches where we fit hypsometric curves, we resort to rectangular fits at only 3,850 reaches (3.0%), the majority of them being located in river deltas where SWOT measurements feature reduced reliability (Supplementary Information section 5).

After generating a piecewise hypsometric curve, we constrain each paired WSE–width measurement to lie on the curve based on the ratio of assumed measurement error to improve the precision of the observations[37,38]. Because the hypsometric curve represents the true channel bathymetry while accounting for measurement uncertainty, these adjusted observations more accurately reflect the underlying hydrologic conditions as they would appear with minimal random error in the SWOT data. From the hypsometric curves we fit at each reach, we integrate the WSE–width piecewise functions using equation (1) and constrained WSE–width observations to obtain a channel cross-sectional area anomaly $\delta A$ (refs. 37,53) corresponding to each SWOT observation. To facilitate the consistent aggregation of area anomalies across reaches, we perform the integrations relative to the median WSE, which we assume to be equivalent to the mean cross-sectional area[53]. Although this simplification introduces a minor residual error between the actual median area and the area at the median WSE, it enables the generation of cross-sectional area anomalies with a near-zero median (Extended Data Fig. 2c).

We convert the zero-median cross-sectional area anomaly time series $\delta A$ at each reach to an associated river volume change ($\delta V$) by:

$$\delta V = L\delta A \tag{3}$$

where $L$ is the length of each SWORD reach. SWORD reach lengths are spatially variable, but the majority of reach lengths are between 10 km and 20 km (ref. 34). Much like the cross-sectional area anomaly, the resulting $\delta V$ only reflects the variability of storage around the observed median WSE, rather than full storage magnitude. As our quality filtering of SWOT observations creates gaps in the storage anomaly time series, we interpolate $\delta V$ to dates when the reach was observed by SWOT but the corresponding observation was removed during filtering. This ensures that the resulting $\delta V$ at each reach is reflective of the observational cadence of SWOT. A forward-filling approach is adopted when water is 'likely fully ice covered' (ice_clim_f = 2), to maintain a stable $\delta V$ during the winter months (for example, from November to May in the Arctic), and a linear interpolation between the two surrounding valid observations is performed otherwise. Then, to enable the comparison of SWOT-derived $\delta V$ with other datasets, we subtract the mean $\delta V$ value from each interpolated time series, to produce zero-mean RSAs at SWOT overpass dates $t_{SWOT}$ (equation (4) and Extended Data Fig. 2d). Finally, we calculate the monthly mean RSA at each reach (equation (5)), further allowing for the aggregation of anomalies within each of the 61 Pfafstetter[55] basins in MERIT-Basins.

$$RSA(t_{SWOT}) = \delta V(t_{SWOT}) - \overline{\delta V} \tag{4}$$

$$\forall m \in [10/23; 09/24], \quad RSA(m) = \underset{t_{SWOT} \in m}{mean} \{RSA(t_{SWOT})\} \tag{5}$$

We quantify the timing of the maximum RSA (Fig. 2a) and the annual range in river storage variability ($\Delta$RSA) by computing the peak-to-trough amplitude of monthly RSA (equation (6)) at the reach scale (Fig. 2b) and at the basin scale (Fig. 3).

$$\Delta RSA = \underset{m \in [10/23;09/24]}{max} \{RSA(m)\} - \underset{m \in [10/23;09/24]}{min} \{RSA(m)\} \tag{6}$$

It is noted that: (1) reach-scale monthly RSA values are first summed to obtain basin- and global-scale time series, after which $\Delta$RSA is computed at each aggregation level; and (2) the choice of the median WSE as the reference to present the changes in RSA does not affect the reach-, basin- or global-scale seasonal analysis of RSA nor the estimation of river storage variability $\Delta$RSA.

## Uncertainty quantification of RSA from SWOT

The simultaneous measurements of river width and WSE provided by SWOT represent an unprecedented advancement in global hydrology, but their validation remains an ongoing and highly complex effort owing to the mission's global scope and novel data products[40]. We provide reliable, conservative uncertainty estimates using error propagation as described in ref. 10. These storage uncertainties have been shown to be reliable in the context of nadir altimetry- and imagery-derived RSA estimates. The computation is based on the calculation of uncertainties in SWOT-derived cross-sectional area changes ($\delta A$) at a river reach for a given time, which are propagated to the RSA estimates from: (1) the uncertainty associated with the width/elevation pairs; and (2) the uncertainty in the piecewise linear hypsometric regression coefficients. The reader is referred to the supplementary information of ref. 10 for thorough information and assumptions about uncertainty quantification used in the modules of the FLaPE-Byrd library.

Further details, associated with uncertainty, on the observational and methodological limitations of the dataset, as well as on the presence of outliers, are provided in Supplementary Information sections 3 and 5 with supporting refs. 56–61.

## Seasonal validation against global river gauges

As a result of the mission's novelty, fine-scale global coverage and the limited range of dynamics covered so far (with only 1 year of measurements), direct validation of the SWOT-derived RSA time series is not yet genuinely feasible. Nevertheless, indirect consistency checks can be performed by comparing SWOT RSA with in situ river discharge records close to the basin outlet, even though the two variables are only loosely related owing to differences in flow velocity and hydrological response times, and water resource management.

To evaluate this consistency, we rely on global in situ river discharge records from the Global Runoff Data Centre[62] (GRDC), selecting 61 gauges, each carefully and manually chosen to be representative of a distinct SWORD basin. For each pair, we compute the correlation between the SWOT basin-scale RSA and the mean monthly discharge time series (Extended Data Fig. 3). This seasonal validation emphasizes the reliability of SWOT across most regions (equatorial, tropical and mid-latitude basins), while also highlighting its current difficulties in reliably monitoring Arctic rivers (for example, the Lena, Khatanga, Glomma and Thelon). These low correlations in high-latitude basins are consistent with the difficulty in determining the exact extent of a river in snowy and wetland environments from SWOT, and the overall lack of usable measurements over frozen rivers (Extended Data Fig. 1), and suggest limitations of interpolating RSA during the winter months. Discrepancies in basins such as the Paraná or the Colorado could indicate limitations in the current ability of SWOT to consistently delineate

water extent in complex and heavily managed hydrological settings. This seasonal validation probably also reflects actual hydrologic and hydrographic mismatches between variations measured in river water storage at a basin scale and in river discharge at a representative gauge.

### Hydrography translations

Details on the Mean Discharge Runoff and Storage (MeanDRS)[3] dataset, used in this study as prior knowledge for global river water storage change comparison, can be found in Supplementary Information section 1, with supporting refs. 63–67. As SWOT observations are made along SWORD reaches, whereas the MeanDRS simulations are made along MERIT-Basins reaches, the fundamental differences between the two hydrographic networks pose challenges for directly comparing river storage estimates. To overcome these differences, we leverage translations between individual reaches in MERIT-Basins and SWORD from the MERIT–SWORD dataset[45] to facilitate the transfer of hydrologic information between networks. We use translations from the MERIT–SWORD dataset to identify reaches in MERIT-Basins (and therefore, storage simulations in MeanDRS) that directly correspond to SWOT-observed reaches in SWORD. For each SWORD reach where we compute RSA from SWOT, we retrieve the associated MERIT-Basins reaches from MERIT–SWORD and store the degree of overlap between the associated reaches. This subset of selected reaches from MERIT-Basins, and their associated storage time series from Mean-DRS, represents a benchmark for comparison with SWOT-derived RSA estimates.

### Comparison with global simulations

To enable a consistent comparison between SWOT observations and MeanDRS simulations, we compute the total MeanDRS RSA time series for each basin by aggregating the storages from all corresponding MERIT-Basins reaches over the 30-year simulation period and across the 3 residence time scenarios. When a MERIT-Basins reach only partially overlaps with a SWORD reach, we apply a weighting based on the fractional overlap of reaches provided by the MERIT–SWORD dataset. This ensures that only the storage from the portion of the MERIT-Basins reach that corresponds to an observed SWORD reach is included. As MERIT-Basins reaches are generally more sinuous and therefore longer than SWORD reaches, we apply a global scaling factor during the weighting process. This factor, equal to 1.13 and calculated from the ratio of MERIT-Basins to SWORD reach lengths, helps to more accurately represent the actual extent of overlap between the two networks[45]. In rare cases where no equivalent MERIT-Basins reach exists for a given observed SWORD reach, we exclude the storage obtained from SWOT for that reach from comparison.

We calculate the RSA for each of the 3 residence time scenarios by subtracting the 30-year mean from the retrieved MeanDRS river storage time series corresponding to the observed SWORD reaches. As the original MeanDRS dataset estimates river storage at all approximately 3 million MERIT-Basins reaches, we note that the variability of the Mean-DRS storage anomalies that we calculate is smaller in magnitude than those reported in ref. 3. To enable comparison with SWOT-observed storages, we summarize the MeanDRS RSA time series by calculating global monthly means and standard deviations over the 30-year period. These metrics characterize the typical magnitude and variability of anomalies for each month, allowing for a direct comparison between SWOT-derived monthly RSA values at observed reaches and the modelled storages from MeanDRS in a typical year (Fig. 4a).

Regional hydroclimatic variability (for example, El Niño periods) exerts a substantial influence on comparisons between a single year of SWOT observations and long-term gauge-corrected simulated means, as the study period captured by SWOT (2023–2024) may represent conditions that are substantially different from climatological averages. The exceptional drought in the Amazon Basin since 2023, with record lows observed in several major rivers including the Negro,

which hit century-low levels[16,47], probably resulted in SWOT-observed variability for 2023–2024 being much lower than that of the 30-year MeanDRS average. To explore whether the storage anomaly derived from SWOT observations aligns with specific historical years in the MeanDRS record, we compare the SWOT-derived RSA to the 30 individual annual time slices of MeanDRS storage simulations across all 3 residence time scenarios (Fig. 4b and Extended Data Figs. 4 and 5). Although the SWOT-derived RSA is generally lower in magnitude than the simulated anomalies in each scenario, we observe the closest agreement during the least variable, and probably driest, years within the historical record for the lowest-volume MeanDRS scenario (Fig. 4b). This suggests that either the period of SWOT observations used in this study coincided with unusually low global river storage variability, or, alternatively, that even the lowest-volume MeanDRS scenario may overestimate the true variability in river storage. The persistent divergence between SWOT observations and the normal and high-volume MeanDRS scenarios, even during years of minimal variability, could further indicate that the land run-off inputs and residence time assumptions underlying these scenarios may not be valid at the global scale. In particular, it is possible that the upper bound for celerity in the model runs ($1.4\ \mathrm{m\ s^{-1}}$), underlying the low-volume scenario (Fig. 4b), remains too low, or that assuming spatiotemporally constant global celerity values is overly unrealistic.

### Quantifying agreement with simulations

To assess spatial variations in the agreement between SWOT and MeanDRS RSA, we compare the Pfafstetter basins with the largest SWOT-observed river storage variability (Extended Data Fig. 6). The Amazon River Basin and the Yenisei and Lena River basins show some of the largest differences between SWOT and MeanDRS, probably owing to a record drought in the Amazon and challenges in resolving partially to fully frozen Arctic rivers from SWOT, respectively. The Nile River Basin features the most striking discrepancy with a SWOT-observed ΔRSA of only $8.5 \pm 1.6\ \mathrm{km^3}$ in 2023–2024, much lower than previous knowledge estimates ($93\ \mathrm{km^3}$ for the MeanDRS low-volume scenario; $160\ \mathrm{km^3}$ referenced in ref. 2). We note that the MeanDRS simulations over the Nile Basin were not bias-corrected owing to the lack of long-term in situ observations and appear to overestimate river discharge by a factor of 3 to 6 (compared with older in situ data from the GRDC), and therefore overestimate storage variations accordingly. With more SWOT observations spanning full annual cycles (when version D products are available) and more local validation endeavours, clearer explanations will emerge.

We also compute several metrics of difference between the 2 volume estimates in each of the 61 Pfafstetter basins. First, we determine the ratio of ΔRSA between SWOT and the MeanDRS low-volume scenario, which shows the highest global agreement with SWOT (Extended Data Fig. 7). In addition, we compute this variability ratio for each of the 3 MeanDRS residence time scenarios and identify the scenario with a ratio closest to 1 for each basin, indicating the best agreement with SWOT (Extended Data Fig. 8).

We evaluate the temporal alignment between SWOT and MeanDRS RSA by calculating both unlagged and circularly lagged Pearson correlation coefficients. We compute the unlagged Pearson correlation between the SWOT and MeanDRS RSA time series within each basin to evaluate their agreement in time (Extended Data Fig. 9). We also perform a circular lag analysis by incrementally shifting the MeanDRS time series by 1 month and recalculating the correlation at each step, covering all 12 possible monthly lags. This approach quantifies the extent to which the SWOT and MeanDRS volume anomalies may be temporally offset. For each region, we identify and plot the lag that yields the highest correlation, indicating the best temporal alignment (Extended Data Fig. 10). Correlations between SWOT-derived RSA and both representative in situ discharge (Extended Data Fig. 3) and Mean-DRS RSA (Extended Data Fig. 9) show strong and regionally consistent

agreement. We note that lags between SWOT and MeanDRS RSA time series can also be the result of monthly lumped routing in MeanDRS, where run-off is accumulated from upstream to downstream without accounting for horizontal travel time from land to rivers or within the river system.

## Data availability

SWOT input data, SWOT-derived active river corridor shape and RSA data are made publicly available via Zenodo at https://doi.org/10.5281/zenodo.18344109 (ref. 68). The SWOT River Database (SWORD) v16 is available at https://doi.org/10.5281/zenodo.10013982 (ref. 69). The MeanDRS[3] model simulations used in this study can be accessed at https://doi.org/10.5281/zenodo.8248069 (ref. 70). The MERIT–SWORD translational dataset[45] can be found at https://doi.org/10.5281/zenodo.13152825 (ref. 71). The river gauge data were downloaded from the Global Runoff Data Centre[62] (GRDC) website, and are available via Zenodo at https://doi.org/10.5281/zenodo.18344109 (ref. 68). Supplementary Figs. 6–8 use © 2026 Google Satellite Imagery as basemaps, for visualization purposes only. We follow the NASA Open Science guidelines for our open-science practices.

## Code availability

The software used in our analysis is available via GitHub at https://github.com/jswade/SWOT-river-volume, and via Zenodo at https://doi.org/10.5281/zenodo.18344109 (ref. 68). We follow the NASA Open Science guidelines for our open-science practices.

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

**Acknowledgements** A.C., J.W., C.H.D. and R.P.M.F. were supported by the Jet Propulsion Laboratory, California Institute of Technology, under a contract with the National Aeronautics and Space Administration (NASA) including grants from NASA's Earth Science US Participating Investigator (NNH20ZDA001N-EUSPI), Earth Science Applications: Water Resources (NNH21ZDA001N-WATER), and the SWOT Science Team (NNH23ZDA001N-SWOTST) programmes. T.P. was supported by NASA Grant 80NSSC25K7715. H.O. was supported by the CNES SWOT TOSCA programme.

**Author contributions** Conceptualization: C.H.D., J.W. and A.C. Methodology: J.W., M.D., A.C., C.H.D. and R.P.M.F. Formal analysis and investigation: A.C., J.W., C.H.D., M.D., R.P.M.F., T.P. and H.O. Data curation: J.W. and A.C. Writing—original draft: A.C., J.W. and C.H.D. Writing—review and editing: A.C., J.W., C.H.D., M.D., R.P.M.F., T.P. and H.O. Visualization: A.C. and J.W. Project administration and financing acquisition: C.H.D. and R.P.M.F.

**Competing interests** The authors declare no competing interests.

**Additional information**
**Correspondence and requests for materials** should be addressed to A. Cerbelaud or C. H. David.

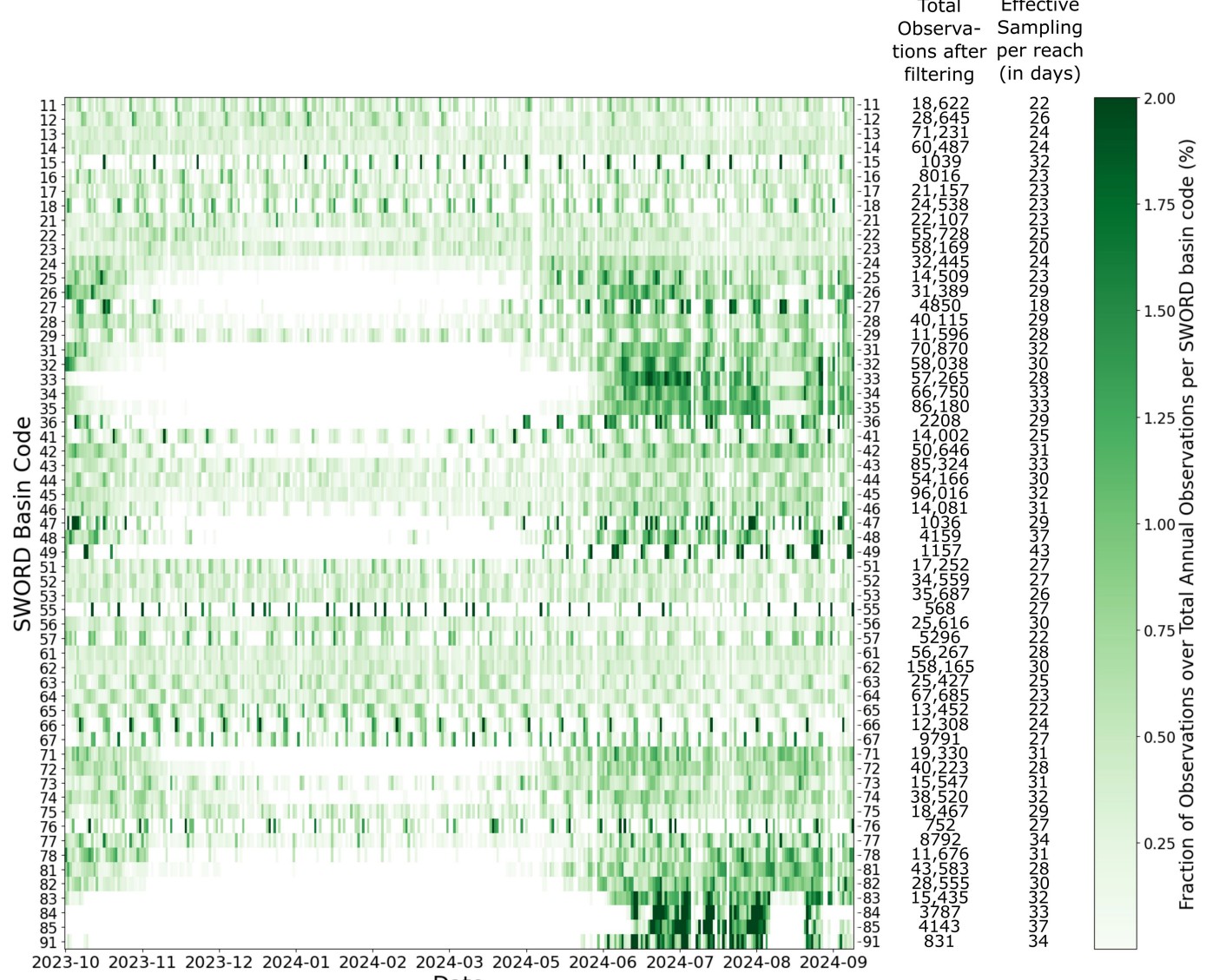

**Extended Data Fig. 1 | Distribution of SWOT measurements retained in this study (i.e., after filtering).** Measurements cover 126,674 river reaches from October 2023 to September 2024 and are given by SWORD basin code. No measurements are processed over rivers affected by ice cover, typically from November to May in Arctic rivers. Regions of Mongolia, North-Western China, Tibet, the Himalayas, i.e., complex mountainous terrains and high-altitude cold regions, also lack post-filtering data typically from November to May.

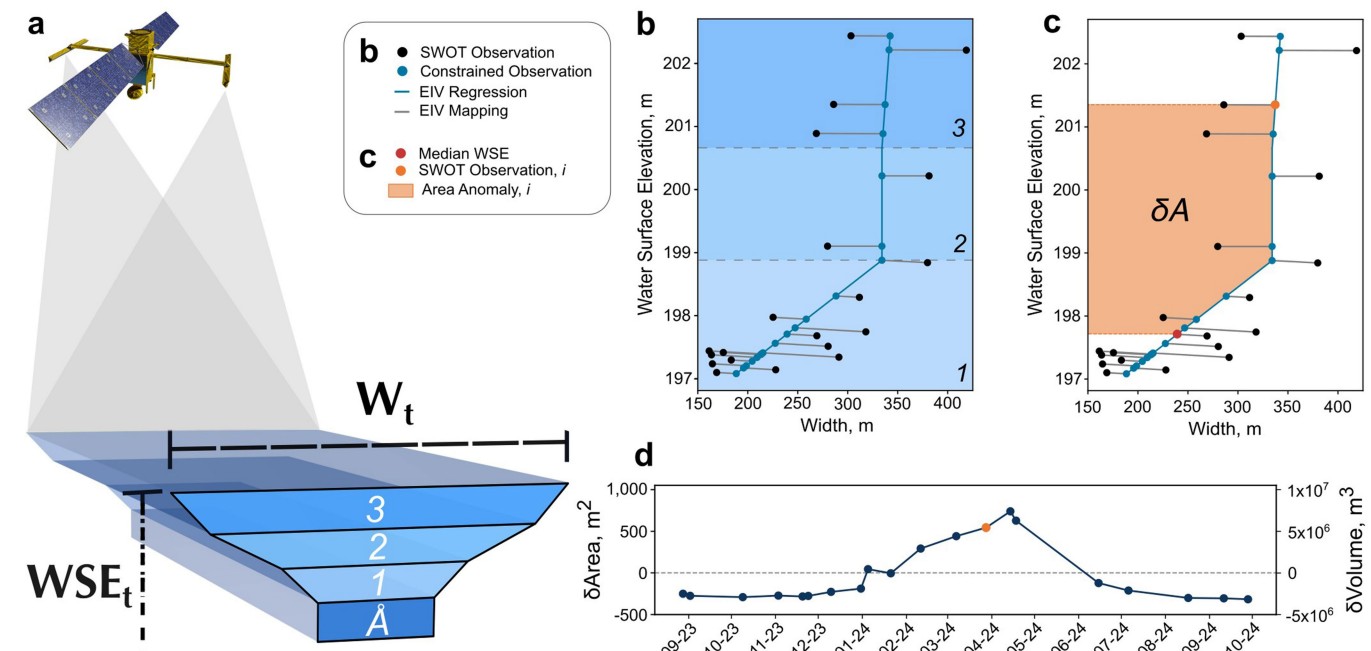

**Extended Data Fig. 2 | Conceptual diagram of SWOT-derived active riverbed shape and river storage anomalies. a**, W and WSE observations and assumed active channel bathymetry for a given reach. $\AA$ corresponds to the unobserved cross-sectional area below the lowest observed WSE. **b**, Elevation-width relationships built using "errors-in-variable" (EIV) with three river corridor zones. **c**, Reach cross-sectional area anomaly $\delta A$ computation relative the median WSE. **d**, Time series of zero-mean reach river storage anomaly $\delta V$. Not to scale. Compared to Fig. 1, river width is plotted in panels b and c along the x-axis and not centered around the vertical axis. Credit: SWOT spacecraft in **a**, courtesy of NASA/JPL-Caltech.

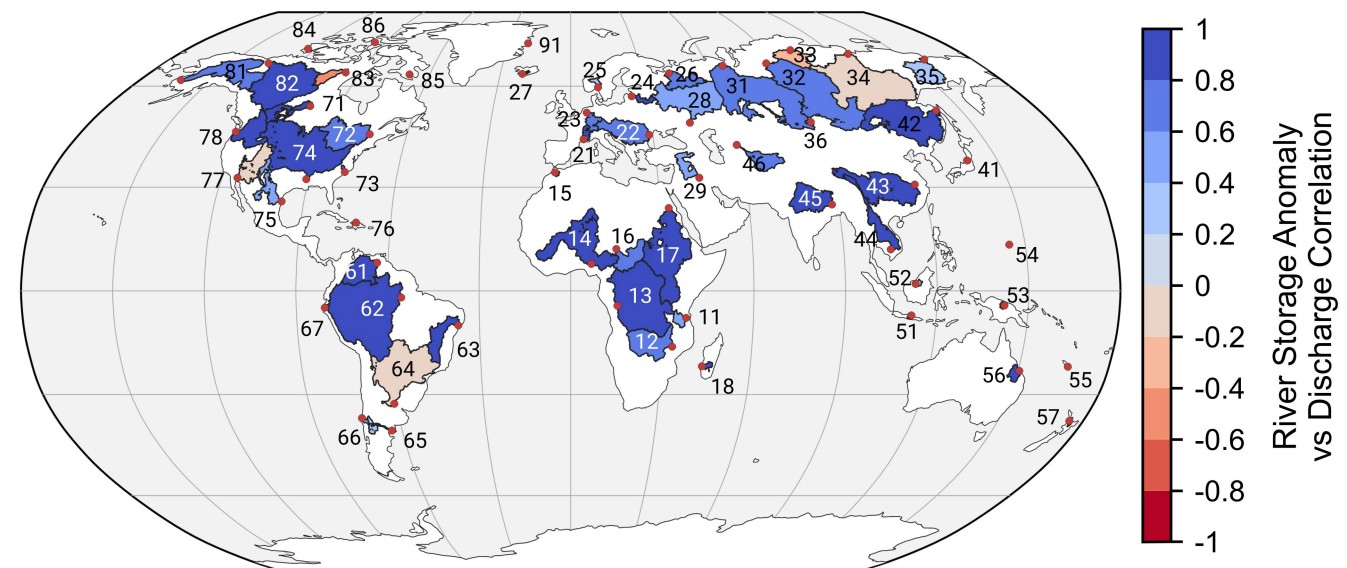

**South America**

| SWORD Basin Code | GRDC River name | GRDC Gauge ID | |
|---|---|---|---|
| 62 | Amazonas | 3629001 | 0.85 |
| 61 | Orinoco | 3206720 | 0.85 |
| 64 | Rio Parana | 3265601 | -0.03 |
| 63 | Rio Sao Francisco | 3651900 | 0.87 |
| 66 | Rio Biobio | 3179500 | 0.59 |
| 65 | Rio Negro | 3275990 | 0.35 |
| 67 | Rio Chira | 3947100 | -0.34 |

**Central and East Asia**

| SWORD Basin Code | GRDC River name | GRDC Gauge ID | |
|---|---|---|---|
| 45 | Ganges | 2846800 | 0.95 |
| 43 | Yangtze | 2181900 | 0.88 |
| 44 | Mekong | 2569003 | 0.95 |
| 42 | Amur | 2906901 | 0.82 |
| 46 | Amu Darya | 2817100 | 0.77 |
| 41 | Agano-Gawa | 2589550 | 0.24 |
| 48 | NA | NA | |
| 47 | NA | NA | |
| 49 | NA | NA | |
| 36 | Khovd Gol | 2744200 | 0.79 |

**Africa**

| SWORD Basin Code | GRDC River name | GRDC Gauge ID | |
|---|---|---|---|
| 13 | Congo | 1147010 | 0.88 |
| 14 | Niger | 1834101 | 0.95 |
| 17 | Nile | 1362600 | 0.90 |
| 12 | Zambezi | 1891500 | 0.78 |
| 18 | Mangoky | 1389090 | 0.95 |
| 16 | Chari | 1537100 | 0.74 |
| 11 | Rufiji | 1286900 | 0.41 |
| 15 | Oued Sbou | 1309700 | 0.54 |

**North and Central America**

| SWORD Basin Code | GRDC River name | GRDC Gauge ID | |
|---|---|---|---|
| 74 | Mississippi | 4127800 | 0.92 |
| 72 | Saint Lawrence | 4243151 | 0.70 |
| 75 | Rio Bravo/Grande | 4351900 | 0.51 |
| 71 | Saskatchewan | 4213550 | 0.91 |
| 73 | Cape Fear | 4148232 | 0.70 |
| 78 | Columbia | 4115201 | 0.92 |
| 77 | Colorado | 4352100 | -0.10 |
| 76 | Rio Yaque Del Norte | 4382100 | -0.15 |

**Arctic (Siberia)**

| SWORD Basin Code | GRDC River name | GRDC Gauge ID | |
|---|---|---|---|
| 31 | Ob | 2912600 | 0.65 |
| 34 | Lena | 2903420 | -0.06 |
| 32 | Yenisei | 2909150 | 0.63 |
| 33 | Khatanga-Kotuy | 2999850 | -0.26 |
| 35 | Kolyma | 2998510 | 0.38 |

**Arctic (America), Greenland and Iceland**

| SWORD Basin Code | GRDC River name | GRDC Gauge ID | |
|---|---|---|---|
| 81 | Yukon | 4103200 | 0.65 |
| 82 | Mackenzie | 4208025 | 0.82 |
| 83 | Thelon | 4214060 | -0.41 |
| 85 | Sylvia Grinnell | 4212700 | -0.53 |
| 27 | Thjorsa | 6401120 | -0.58 |
| 84 | Big | 4210450 | 0.13 |
| 91 | Zackenberg | 6998400 | -0.26 |
| 86 | Allen | 4211310 | |

**Europe and Middle East**

| SWORD Basin Code | GRDC River name | GRDC Gauge ID | |
|---|---|---|---|
| 26 | N. Dvina | 6970250 | 0.72 |
| 28 | Volga | 6977100 | 0.60 |
| 29 | Euphrates | 2595400 | 0.52 |
| 22 | Danube | 6742900 | 0.61 |
| 24 | Daugava | 6373307 | 0.85 |
| 23 | Rhine | 6435060 | 0.79 |
| 21 | Rhone | 6139100 | 0.87 |
| 25 | Glomma | 6729403 | 0.30 |

**Oceania**

| SWORD Basin Code | GRDC River name | GRDC Gauge ID | |
|---|---|---|---|
| 56 | Fitzroy | 5101300 | 0.87 |
| 53 | Sepik | 5550500 | 0.35 |
| 52 | Sungai Rajang | 5230300 | 0.35 |
| 51 | Bengawan Solo | 5141200 | 0.83 |
| 57 | Waikato | 5865300 | 0.78 |
| 55 | Tontouta | 5762500 | 0.71 |
| 54 | Ylig | 5974500 | |

**Extended Data Fig. 3 | Comparison of global SWOT-observed river storage anomalies with in situ river measurements.** One representative in situ gauge from the Global Runoff Data Centre[62] (GRDC) is used per basin (61 SWORD basins). GRDC river gauge locations are marked with red dots, and their corresponding drainage areas are color-coded to reflect the correlation between monthly gauge discharge seasonality and SWOT RSA at the basin scale. Each validation gauge is labeled with its associated SWORD basin. A summary table presents these correlation values by world region and SWORD basin (ranked by descending SWOT RSA). Basemap from Natural Earth (https://www.naturalearthdata.com).

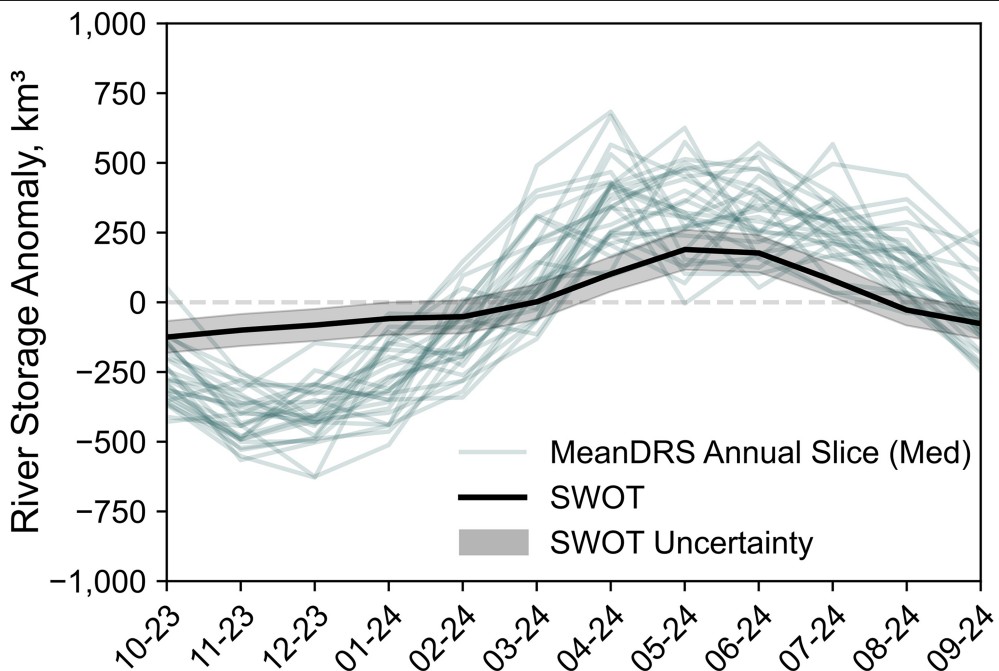

**Extended Data Fig. 4 | Comparison of global SWOT-observed river storage anomaly with 30 years of model simulations, at equivalent reaches.** Colored annual slices represent MeanDRS[3] model medium volume scenario, aligned to SWOT observation period. The shaded region represents the SWOT uncertainty envelope.

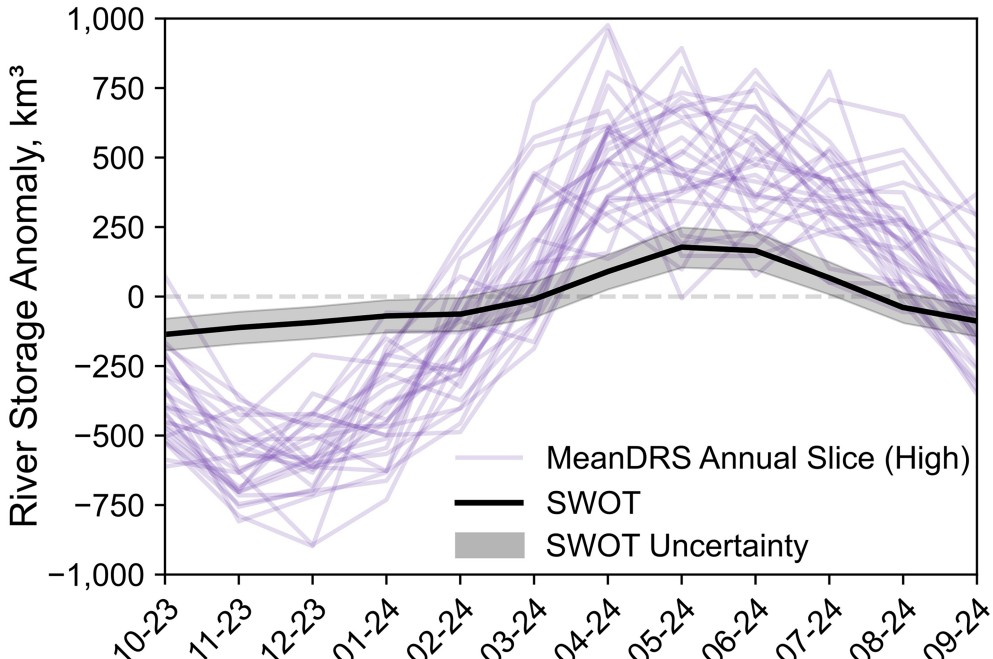

**Extended Data Fig. 5 | Comparison of global SWOT-observed river storage anomaly with 30 years of model simulations, at equivalent reaches.** Colored annual slices represent MeanDRS[3] model high volume scenario, aligned to SWOT observation period. The shaded region represents the SWOT uncertainty envelope.

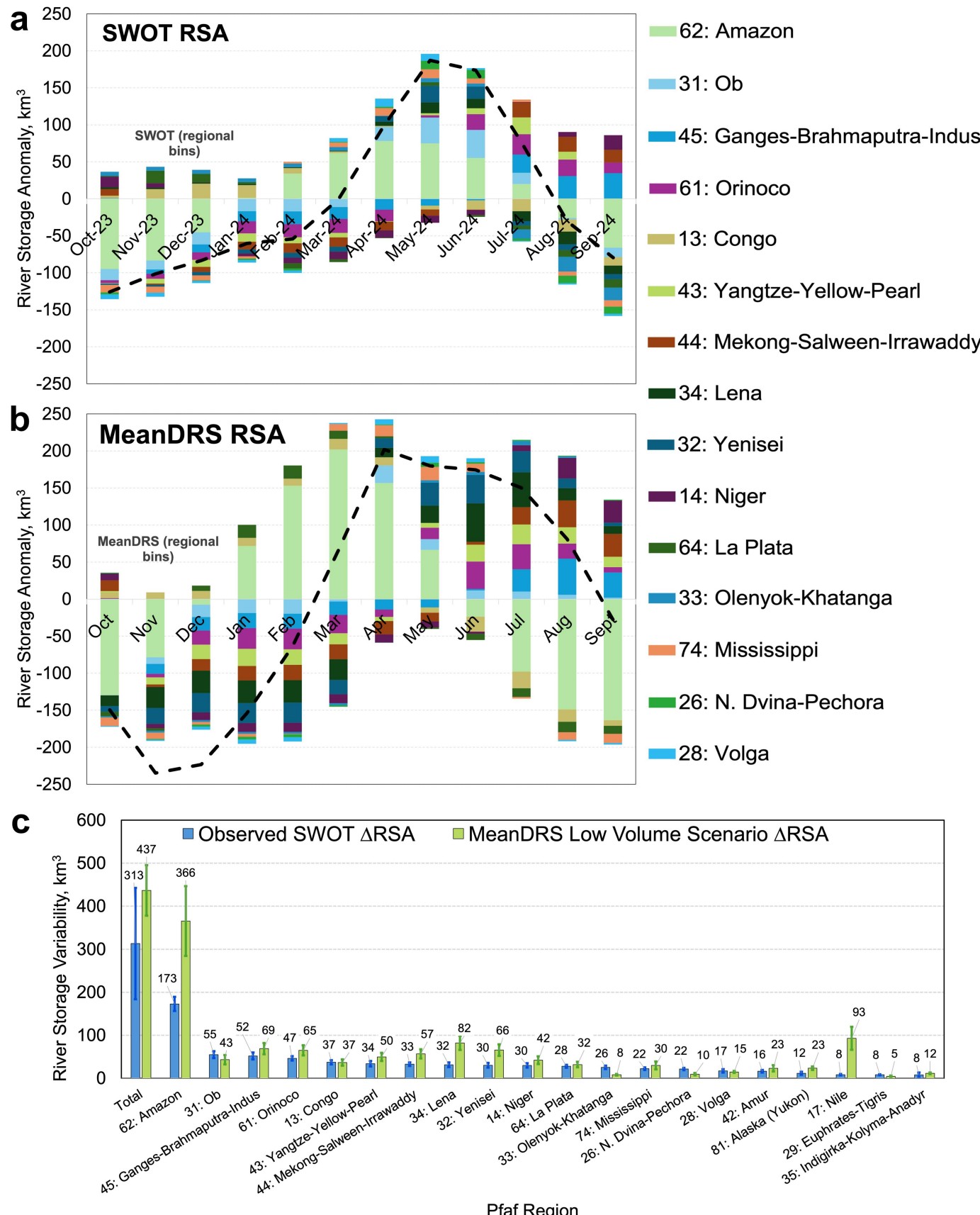

**Extended Data Fig. 6 | Regional SWOT-observed and modeled river storage anomaly at equivalent reaches for the basins with the largest river storage variability.** Modeled storage anomaly correspond to MeanDRS[3] low volume scenario. **a**, Regional monthly contribution to SWOT global RSA from October 2023 to September 2024. **b**, Regional monthly contribution to MeanDRS global RSA (30-year mean). **c**, River storage variability (ΔRSA). ΔRSA is calculated as the annual range in monthly storage anomalies. Vertical error bars reflect the uncertainty estimates.

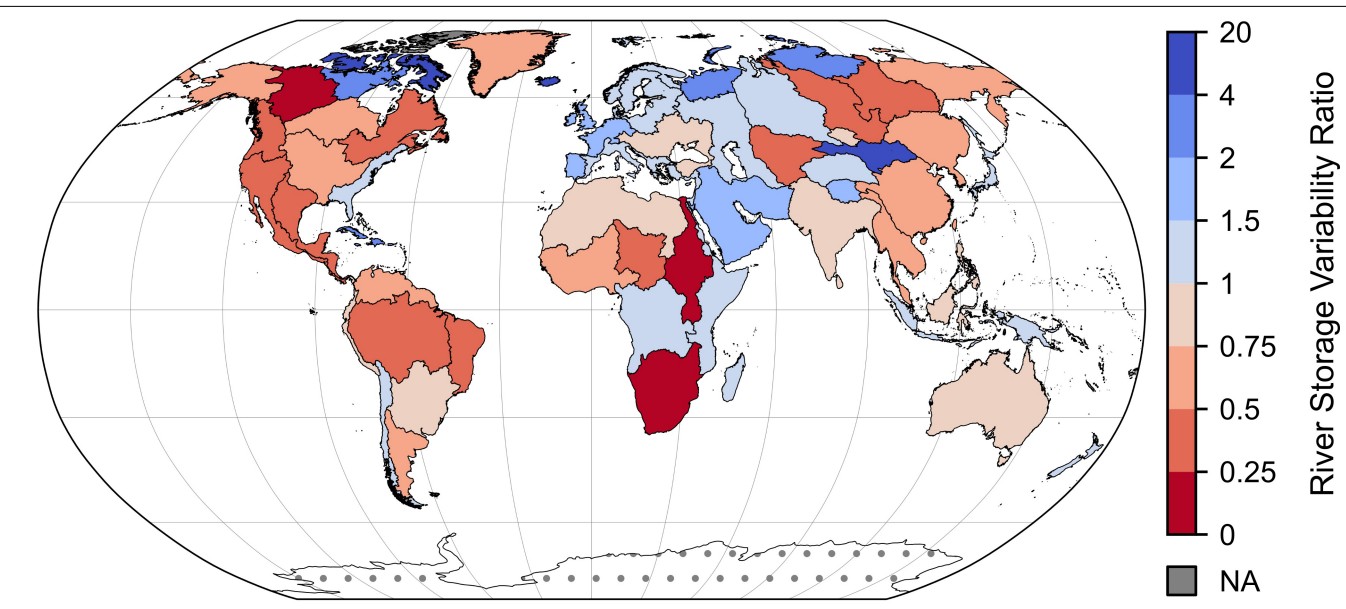

**Extended Data Fig. 7 | Regional ratio of river storage variability between SWOT-observed and modeled storage anomaly, at equivalent reaches.** Modeled storage anomaly correspond to MeanDRS[3] low volume scenario. ΔRSA is calculated as the annual range in monthly storage anomalies. NA, excludes Antarctica. Basemap from Natural Earth (https://www.naturalearthdata.com).

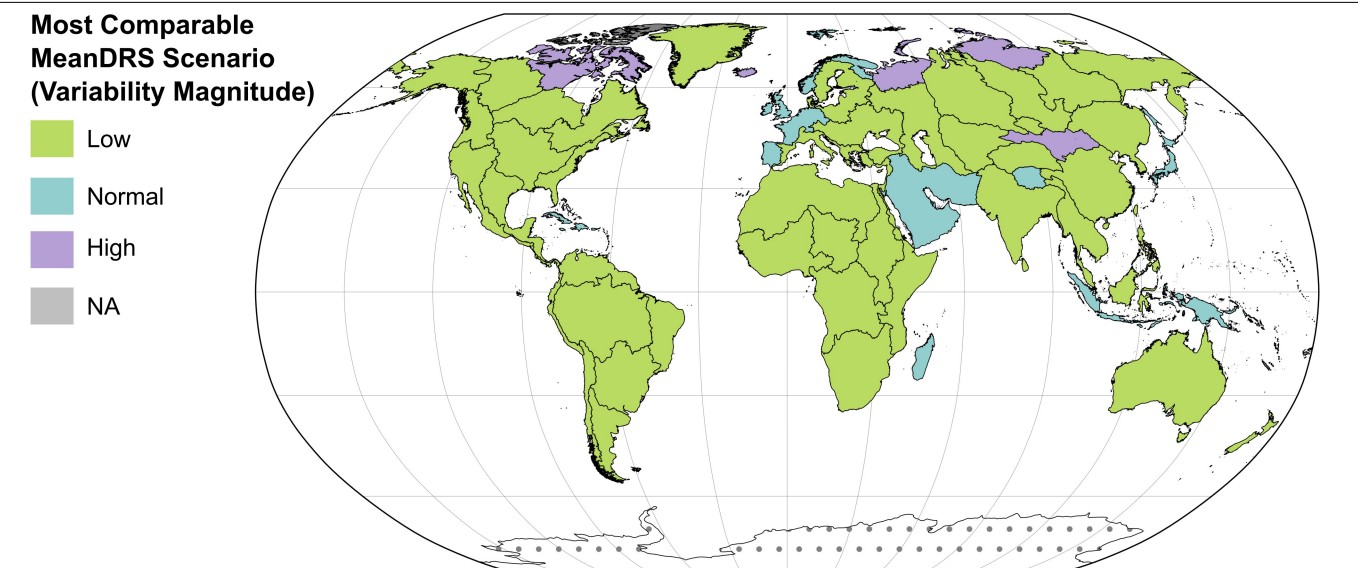

**Most Comparable MeanDRS Scenario (Variability Magnitude)**

- Low
- Normal
- High
- NA

**Extended Data Fig. 8 | Most comparable model volume scenario based on regional river storage variability ratio between SWOT-observed and modeled storage anomalies.** The three model volume scenarios correspond to MeanDRS[3] low, medium (or normal), and high volume simulations. ΔRSA is calculated as the annual range in monthly storage anomalies. NA, excludes Antarctica. Basemap from Natural Earth (https://www.naturalearthdata.com).

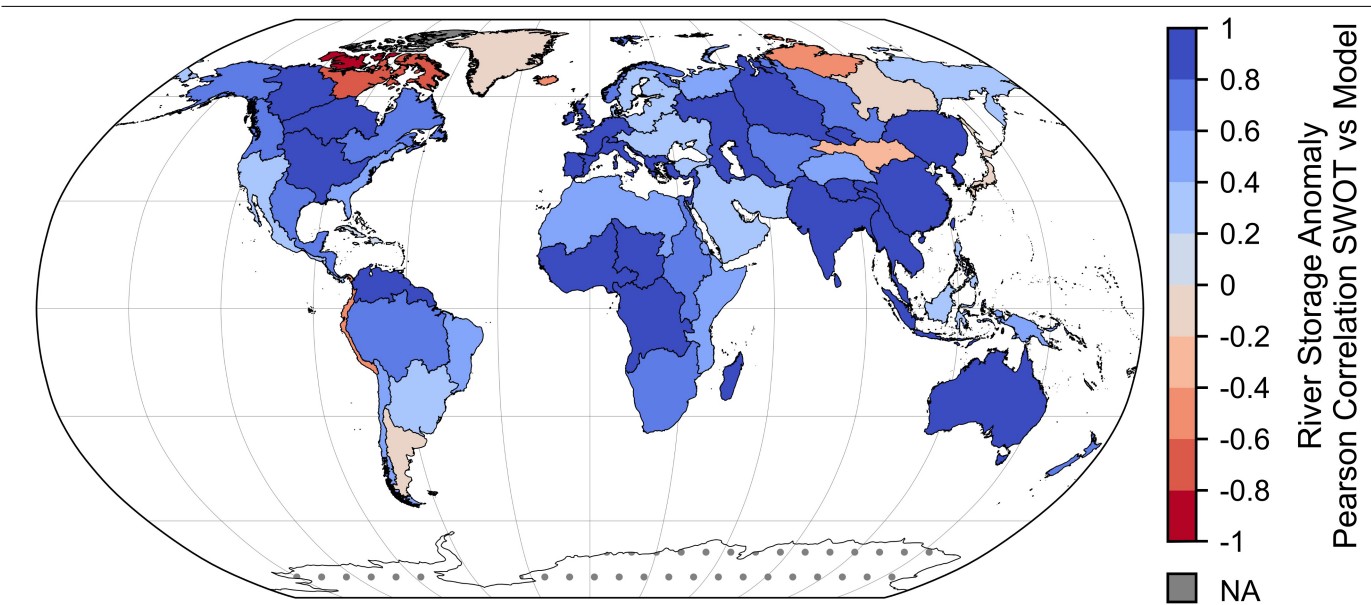

**Extended Data Fig. 9 | Pearson correlation coefficient between regional SWOT-observed and modeled river storage anomalies, at equivalent reaches.** Modeled storage anomaly correspond to MeanDRS[3] low volume scenario. NA, excludes Antarctica. Basemap from Natural Earth (https://www.naturalearthdata.com).

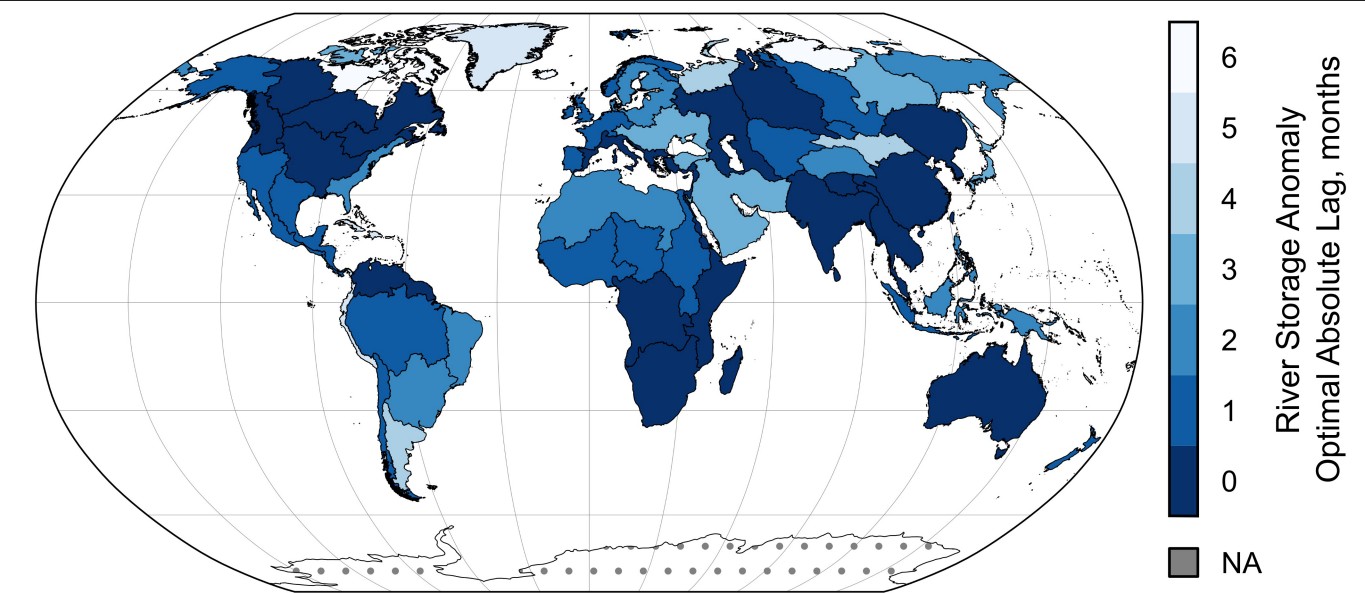

**Extended Data Fig. 10 | Optimal temporal lag between regional SWOT-observed and modeled river storage anomalies, at equivalent reaches.** Modeled storage anomaly correspond to MeanDRS[3] low volume scenario. Optimal lag calculation is based on Pearson correlation. Forward and backward temporal lag treated as equivalent. NA, excludes Antarctica. Basemap from Natural Earth (https://www.naturalearthdata.com).

**Extended Data Table 1 | Number of unique river reaches (first line) and observations (second line) from successive data quality filtering of SWOT data**

| Reaches SWOT files | rch_type = 1 or 5 | reach_q < 3 | xovr_cal _q = 0 | dark_fra c ≤ 0.3 | ice_clim_ f = 0 | obs_frac _n ≥ 0.5 | \|xtrk_dis t\| ∈ [10-60] km | Reaches with ≥ 5 obs | WSE range < 20 m |
|---|---|---|---|---|---|---|---|---|---|
| **215,485** | 171,824 | 170,791 | 170,675 | 170,079 | 169,914 | 164,105 | 158,176 | 131,344 | **126,674** |
| **9,948,845** | 8,143,747 | 7,889,505 | 4,746,552 | 3,527,809 | 2,905,972 | 2,076,785 | 1,788,292 | 1,716,718 | **1,646,813** |

Valid reach-level observations from the SWOT Level 2 KaRIn High-Rate River Single Pass (RiverSP, version C PIC0/PGC0) between 1 October 2023 and 30 September 2024.