## [Peer Review File · Nature]

Wide-swath Altimetry Maps Bank Shapes and Storage Changes in Global Rivers

Corresponding Author: Dr Arnaud Cerbelaud

Version 0:

Reviewer comments:

Referee #1

(Remarks to the Author)

This paper leverages new data from the SWOT mission over a water year to quantify variations in water storage in rivers at a monthly scale. The work also cleverly utilizes these data to detect the shape of river banks and thus the variations in cross section in the active portion of rivers as water level varies. The authors also compare their results with previous storage estimates obtained from modeling.

Exciting analysis and paper. This is the type of study I was really looking forward to seeing with SWOT data—the opportunities these data offer for analyzing the global hydrologic cycle are numerous, and this is an exciting first glimpse at what a full water year from SWOT looks like.

I very much like the idea behind this paper, as well as the approach and the metrics used to quantify variations in storage in the world's rivers and the comparison to the modeling study. I also found the concept of tracking water levels from low to high and thus the shape of the active channel particularly exciting. Combined with water extent, water level measurements allow for measurements of river shape from the lowest to the highest water surface elevation detected by SWOT. Super cool.

However, I do have a major set of questions related to the SWORD centerlines and the potential impact of their inaccuracies on the study presented. These questions are outlined below:

- Did you perform any selection of the reaches, or were all SWORD reaches included? From what is written in the paper, it doesn't seem like a selection was made. I understand there was some filtering based on flags on the water surface elevation, but it's unclear whether any checks on centerline quality were performed. I ask because I've used SWORD extensively and have retraced centerlines in many global river deltas, where SWORD centerlines were often quite inaccurate. I've also examined centerlines in other rivers I work on and, in some cases, observed centerlines placed in the floodplain (e.g., the meandering Trinity River has several such reaches).

- Even if the centerlines were perfect—which they are not—some rivers are extremely dynamic. The Ganges is mentioned frequently in the paper, and that system is highly dynamic, with SWORD centerlines that are notably off.

- A related and more specific question concerns coastal areas: were delta reaches included, or did the study stop upstream of deltas? If deltas were included, were variations due to tides accounted for? If so, it might be worth analyzing those reaches separately from the rest of the basin.

I understand this is a global-scale study and that checking the accuracy of every reach is not feasible. However, I would suggest double-checking especially those reaches that show extreme (either large or small) variations in storage. The small variations, in particular, could be cases in which the centerline is missing the channel altogether.

This is a critical point, as the quality of the centerline potentially affects all analyses performed with the data. While the authors are likely aware of these issues, the current information in the paper did not reassure me about how inaccuracies in SWOT might propagate through the analysis and affect the storage and related metrics and how the authors may have accounted for these issues in their analysis.

My suggestion is to include some evidence in the paper that centerlines were checked, particularly for cases where extreme storage variations were observed. Additional figures with insets showing the Ganges and other systems could help illustrate where the centerlines fall relative to imagery and/or the SWOT water mask in raster form.

Other comments/questions:

Line 60: I understand that centerlines were traced for rivers wider than 30 m, but we don't trust SWOT for rivers narrower than 50 m, correct?

Line 91: This is not exactly the river shape, but rather—as specified here—the river shape between the lowest and highest water surface elevations. Unless these are ephemeral rivers, this represents only a portion of the channel shape. I missed this earlier but effectively you called it the active channel shape – it might be helpful to include this definition in Extended Figure 2 (at least in the caption) so that terminology is consistent.

Line 105: I understand these two systems may have comparable discharge, but one is covered with permafrost and the other is not. I wonder if comparing systems within the same climatic zone might be more meaningful.

Figure 1: I suggest changing the line style for the median—it's currently very hard to see, which makes interpreting the figure difficult. For example, in the Ganges, the mean is outside the IQR bounds, suggesting some extreme values are shifting the mean. Where did those come from? And is the median underneath the mean?

Figure 2: This is a very important figure, but it's hard to see anything at this scale beyond overall color and variability (unless that is what you wanted to show). I suggest creating additional insets that show major river basins in more detail so that centerlines can be seen and analyzed—especially for Figure 2a.

Apologies if I missed anything and I do hope these comments are helpful!

Paola Passalacqua

(Remarks on code availability)

I did not install the code and only had a look at the github page to see the associated documentation, which is comprehensive.

Referee #2

(Remarks to the Author)

The SWOT satellite is designed to observe Earth's surface waters and (sub-)mesoscale ocean circulation, employing a novel Ka-band Radar Interferometer for the first time. This paper focuses on mapping riverbank shapes and water storage changes of global rivers using SWOT, which are also the main goals of SWOT. It is interesting to investigate and reveal the river changes and their characteristics in different basin scales and the global scale. Although this study provides the first near-global measurements of active river corridors and river storage variability, I have some suggestions and concerns about SWOT and the results.

Concerns about the goal of SWOT:

Lines 427-438. After a series of filtering, 126,674 river reaches are retained from ~240,000 global river reaches. More details should be given on how many reaches are filtered at each step, and the reasons for their exclusion based on different quality criteria or conditions. It would also be valuable to analyze the extent to which SWOT has achieved its intended goals for global river observations.

Lines 480-482. The scientific requirements for river width and WSE from SWOT are 30 m and 0.1 m, respectively. According to the Science Requirements Document, one of SWOT's requirements is to map rivers wider than 100 m, with an extended goal of 50 m. Have the requirements been validated with other technologies and measurements? Could the uncertainties of these two variables be used to provide uncertainty in storage changes? In the input data provided by the authors, I examined three basins (Amazon, Yangtze, and Ganges) and found that the typical median WSE uncertainty (wse_u) is about 0.4~0.6 m, and ~0.17 m for rivers 50~100 m and >100 m wide. Do these results indicate that SWOT has actually met its stated requirements and goals? If not, does this imply that the estimates for smaller rivers may be inaccurate?

Line 733, Extended Data Figure 1. Are all the rivers without measurements from November to May located in the Arctic? How about rivers in complex mountainous terrains and high-altitude cold regions, such as the Tibetan Plateau? Can you give the distribution or analysis of the rivers that have more missing data?

Concerns about the processing and analysis of river activities:

Line 100, Figure 1. The interquartile ranges of global rivers vary significantly. For example, for Congo River, it could be larger than 1000m, while for the highly active Amazon River, it could be less than 1000m. Is it affected by the SWOT's capabilities and uncertainties?

Line 113. How is frozen water defined, and how to judge whether the observations are over frozen water or not? It is useful to know whether the river observations can be used and are reliable.

Line 119. Could the authors clarify why δA is computed relative to the median WSE, and how the choice of zero reference might affect the RSA results?

Lines 125-127. Which method is used for the interpolation of missing SWOT observations? Since the dates of overpass are irregular and the missing observations vary with the latitude, especially for the rivers over the Arctic area, can the method be applied in these conditions?

Line 132. SWOT provides one valid observation every 28 ± 4.6 days on average, but it varies with the latitude, and the revisit rate is about 10 days. How did you get the monthly variability?

Lines 136-140. Since there is only one year of observation of SWOT, and for each month, there may be only a few observations, is it enough to analyze the maximum river storage for each month? Some extreme river changes may be missed. And the seasonal patterns do not always follow the latitudinal gradients, especially for rivers in the Arctic and the Amur basin. Except for the Amazon River and Congo River basins, how many basins show patterns consistent with current understanding and how many are inconsistent?

Line 168, Extended Data Figures 3. Low correlations also exist in the Rio Parana and Colorado River basins.

Line 461. Formula (1) in the Methods section is theoretically applicable, but for long river reaches or rivers with variable slopes, it may introduce errors. Could the authors clarify how river slope is accounted for in these cases, given that SWOT can, in principle, measure slope?

Lines 491-496. In practice, how do the authors know the width measurements are unreliable, leading to the initial EIV regression failing? Or is it that the initial EIV regression fails, indicating unreliable width measurements? More details should be verified, and the distributions and circumstances of the 3,850 reaches can be concluded.

Line 639. Some basins in Extended Data Fig. 7 exhibit ratios that are unusually large or small, whereas theoretically these ratios should mostly range near 1 (e.g., 0.75–1.5). Could the authors explain the cause of these deviations?

Line 693. In the last part of the Methods, does the term "retracking algorithms" correspond to the same concept as retracking in nadir altimetry?

Concerns about the comparison with MeanDRS:

The authors show that the storage changes of global rivers estimated by SWOT are always lower than the lowest estimates of MeanDRS. They attributed the discrepancy to droughts in the Amazon basin. However, this explanation is insufficient to account for the underestimation observed in many other basins. In addition, considering Amazon's dominant contribution, can the riverbanks and floodplains really be distinguished in dense rainforest regions? Could such potential misidentification lead to severely inaccurate storage change estimates?

Since different basins may exhibit distinct behaviors, performing a global-scale comparison may be too challenging to analyze. Moreover, the global analysis may be affected by the irregular temporal sampling induced occasional missing observations. It would be more reasonable to include an analysis focusing on one or a few representative basins, which could also avoid the influence of seasonal ice cover.

Although the authors studied the correlation of water storage changes between SWOT and gauges, I don't see the estimation of the uncertainties for SWOT- or MeanDRS-derived water storage changes. As a result, it remains uncertain whether SWOT offers advantages compared to current approaches.

Lines 198-199. I can't see this information from Figure 4b.

Lines 202-204. At the annual scale, the Δ RSA in the Amazon basin has the largest impact on the observed global Δ RSA. However, at the monthly scale, what about the RSA in the Amazon basin compared with the global RSA? For example, in Figure 4, which basins primarily account for the discrepancies between SWOT observations and model estimates?

Line 218, Figure 4. There is a larger discrepancy between SWOT RSA and those of the model from November to February than in other months, and there is also a lot of missing data during this period. Any correlation between them?

Most of the time, the hydrological model underestimates the RSA due to incomplete understanding and insufficient runoff and other input data. However, compared with the model's lowest estimate, the SWOT RSA is even lower. What does this mean?

Line 650, Extended Data Figures 10. There appears to be a temporal lag between the SWOT and MeanDRS RSA time series, with some regions showing a lag exceeding three months. Could the authors clarify the underlying cause of this lag?

Extended Data Figures 7–10. Basins in the Mongolia region show results that differ from those of the surrounding basins. Could the authors clarify the reason for these discrepancies?

Overall, the first observed global river bank shape and storage anomaly is derived by the SWOT observations from October 2023 to September 2024, and the river storage changes are compared and verified with the hydrological model estimates. Both annual and monthly basin scale RSAs are emphasized. Beyond the first glance, some new insights could be highlighted. SWOT is capable of mapping rivers wider than 30m at a global scale. However, its temporal coverage is limited, and measurements of narrow rivers (less than 100 m), frozen rivers, or rivers in complex terrains may still be challenging. These limitations may contribute to unexpected patterns in the global- and basin-scale analyses presented in this paper. SWOT has already demonstrated its ability to capture finer-scale ocean dynamics. Therefore, over rivers, is it possible to reveal finer-scale spatial changes, according to different river channel characteristics and surrounding circumstances, especially in typical basins?

(Remarks on code availability)

Several steps (2, 5, 7, 10, and 12) in `tst_pub_repr_all_Wade_etal_2025.sh` failed to run, and outputs using the provided SWOT data did not fully match `output.zip` (e.g., basin 11, Step 1: $m_1 = 68.3843$, `fit_method = simple` vs $m_1 = 65.5352$, `fit_method = set`). More descriptions about the data input and output could be suggested.

Referee #3

(Remarks to the Author)

Review of Manuscript Nature-2025-07-19902: "Wide-Swath Altimetry Maps Bank Shapes and Storage Changes in Global Rivers"

This manuscript presents an analysis of the intra-annual variability in global river water storage using the SWOT satellite observations, providing a straightforward view of river corridor morphology and seasonal storage changes at a global scale. The work uses a critical dataset to address a long-standing challenge in hydrology, and its findings have significant implications for hydrological modeling, water resource management, and climate change studies. The research on global-scale riverbank morphology has the potential to offering a novel perspective of understanding fluvial geomorphology. However, despite its groundbreaking nature, the manuscript in its current form has several fundamental weaknesses that affect the credibility of its core conclusions.

Major comments:

1---A core strength of this research is SWOT's near-global coverage, particularly its novel capability to map riverbank morphology. However, the manuscript does not fully leverage this advantage. The spatial analysis lacks the necessary depth and detail to investigate the heterogeneity of river characteristics across different climate zones (e.g., arid vs. humid, mountainous vs. lowland) and within different sections of river basins (upstream, midstream, downstream). For instance, Figure 1 presents a highly generalized overview by averaging riverbank shapes along the entire mainstream of major rivers, which masks significant spatial variability. The authors should delve deeper into the data to uncover new geographical patterns or hydrological process insights that are only possible with SWOT's high-resolution observations. For example, an analysis of typical bank morphologies in different geomorphic units or climatic settings and their impact on water storage capacity would more effectively demonstrate the value of this new observational tool.

2---The study's reliance on a single hydrological year (2023–2024) to quantify global river storage change is its most critical weakness. An anomalous year, heavily influenced by a historic drought in the Amazon, cannot represent the long-term mean or range of variability for global rivers. This makes it difficult for readers to determine whether the significant discrepancy between SWOT observations and model estimates is due to an underestimation by SWOT, an overestimation by previous models, or simply the effect of this single extreme event. The authors should conduct a more nuanced regional analysis. For the study period, were other major basins (e.g., the Congo, Mississippi) experiencing normal or even flood years? If so, how do SWOT's estimates in those basins compare to models or gauge data? Such regional comparative analysis would help to disentangle the impact of specific extreme events from systematic observational or model biases and provide a more objective assessment of SWOT's capabilities.

3---The study is based on the SWOT River Database (SWORD), but global river systems are heavily regulated by reservoirs, which have a massive impact on river storage variability. The manuscript fails to clarify how reservoirs are handled in the analysis. Were reaches corresponding to reservoirs entirely excluded, or were they treated as natural river segments? Did the authors use an existing global reservoir inventory (e.g., GDW, GeoDAR, GOODD) to precisely identify and separate natural reaches from regulated ones? A clear quantification and distinction of this regulated water volume is essential for an accurate estimation of natural global river storage changes, and this must be detailed in the Methods section.

4---A major weakness of this paper is the lack of a comprehensive quantification of uncertainty in its estimates. Beyond the short study period (mentioned above), uncertainty arises from several other sources:
(a) The reliability of the SWOT product at a global scale is not fully assessed: This is the first global-scale scientific analysis using the SWOT L2 river product. Its performance in estimating river water level and width, particularly its ability to capture

seasonal variability, requires a thorough global validation. The authors are encouraged to perform cross-validation using data from traditional altimetry missions with longer time series (e.g., Jason, Sentinel series) for a selection of representative reaches to assess WSE reliability. Furthermore, the accuracy of SWOT-derived width could be validated against high-resolution Sentinel-1 SAR imagery. This would also help quantify the potential error introduced by SWOT's effective 28-day revisit period in capturing flood processes, especially flood peaks.

(b) The simplified processing for ice-covered rivers is a significant source of error: The manuscript describes a simple "forward-filling" method to handle data gaps during the winter ice-covered season. This approach likely leads to a significant underestimation of storage changes in these regions, as sub-ice flow continues. Given the large number of rivers in the Arctic and on broad mountainous regions/plateaus, the neglected storage variability in these areas could have a non-negligible impact on both the global total and regional patterns.

(c) The reliability of the peak storage month is not evaluated: A key finding presented in Figure 2 is the month in which river storage peaks. However, the robustness of this finding has not been independently verified. It is suggested that the authors validate this result using other multi-source remote sensing datasets (e.g., time series of river width or water level derived from optical or microwave sensors) in representative regions to strengthen the credibility of this conclusion.

Specific comments:

Line 11-25: As specified in the paper title, one of the major contribution of this manuscript lies in the mapping of river bank shapes. However, the findings in such dimension are not described in Abstract at all.

Line 81: The analysis uses data from 126,674 reaches. Could you clarify what percentage this represents of the total number of river reaches (type 1 and 5) in the SWOT v16 database? This would provide better context for the study's spatial coverage from the outset.

Line 88-92: The statement notes that the derived channel shapes are for the bathymetry between the lowest and highest water levels observed. Given that the study period (2023-2024) included a historic drought in the Amazon, which contributes over 50% of the total observed storage variability, the "lowest observed" levels there may be anomalously low. How might this affect the derived bank shapes and their representativeness of a more typical year? What about other basins similarly with hydroclimatic extremes during the study period?

Line 109: The acknowledgment that "floodplains appear to be rarely captured" is critical. Since many global hydrological models explicitly include floodplain storage, could this methodological limitation be a primary driver of the discrepancy between SWOT-observed storage and modeled estimates, rather than solely model parameter errors? This point is understated in the discussion part.

Line 125-127: The manuscript mentions that missing data are infilled. The methods (Line 518) state a "forward filling approach is adopted when water is 'likely fully ice covered'". This assumes storage is static under ice. However, sub-ice flow persists, and storage can change significantly during winter. This assumption will likely lead to an underestimation of storage variability in Arctic and boreal rivers. Has the potential magnitude of this underestimation on the global total been assessed? May this challenging issue for the Arctic rivers be validated with in-situ materials?

Line 155-156: The Amazon basin accounts for 172.7 km³ of the total 313.4 km³ annual river storage variability (ARSA), which is approximately 55%. Given the severity of the 2023-2024 drought, this single basin's anomalous condition heavily influences the global total. It seems problematic to compare this single, anomalous year for the globe's most dominant basin against a 30-year model climatology to draw conclusions about model performance. The paper should more strongly emphasize that the global value is likely not representative of a climatological average.

Line 165-171: The validation uses 61 in situ gauges. Extended Data Fig. 3 shows very low or even negative correlations for several major basins (e.g., Rio Paraná, Colorado, Khatanga-Kotuy). These poor correlations in non-ice-affected, major basins raise concerns about the reliability of the RSA estimates in those regions. Could the authors provide potential explanations for these specific discrepancies?

Line 193-209 & 218-223: in the main text, the comparison between SWOT (10-2023 to 09-2024) and Collins et al. (1980-2009) monthly river storage anomalies was merely made at the global scale. The results may be misleading to audience.

Line 427: The analysis retains SWOT observations flagged as 'suspect' and 'degraded'. What is the rationale for including these lower-quality data points, and how might their inclusion impact the accuracy of the final storage estimates?

Line 435-438: A filter is applied to remove reaches where the observed elevation range exceeds 20 m. This seems like an arbitrary threshold that could systematically exclude some of the world's most dynamic river reaches, particularly in large, low-gradient tropical basins or dam-regulated reaches. Please provide a stronger justification for this filter or conduct a sensitivity analysis to show its impact.

Line 446-448: The effective sampling rate is one observation every 28 days after filtering. This is significantly sparser than the nominal revisit time. As acknowledged in Lines 714-723, such a low sampling frequency is likely to miss the peaks of many flood events, leading to a systematic underestimation of storage variability. This limitation seems significant enough that it should be highlighted more prominently in the main text as a major contributor to the lower-than-modeled storage values.

Line 472-482: The EIV regression uses an assumed WSE uncertainty of 0.1 m and width uncertainty of 30 m based on mission requirements. Since the paper notes the product's error estimates are unreliable, how sensitive are the results (specifically the cross-sectional area anomaly δA) to these assumed uncertainty values?

Line 483-496: The three-part piecewise fitting is used to represent flow regimes, including out-of-bank flow. However, the main text states that floodplains are rarely captured. Please clarify this apparent contradiction. If out-of-bank flow is not being observed, what is the physical interpretation of the third segment of the hypsometric curve, and is there a risk it is fitting to measurement noise rather than a distinct flow regime?

Line 495-502: For the 3% of reaches where a rectangular channel was assumed, this simplification would inherently suppress storage variability (since width does not change with WSE). What is the estimated impact of this simplification on the total global ARSA?

(Remarks on code availability)

Referee #4

(Remarks to the Author)

I co-reviewed this manuscript with one of the reviewers who provided the listed reports.

(Remarks on code availability)

Version 1:

Reviewer comments:

Referee #1

(Remarks to the Author)

Thank you for the opportunity to review the revised version of this manuscript. I compliment the authors for the tremendous work that went into the revisions. The revised manuscript is extremely solid and exciting, and it represents a fundamental scientific advance. It sheds new light on water storage in global rivers and proposes a clever approach to detecting river-bank morphology within the active portions of the channels observed by SWOT.

I was already very excited about the first version of this manuscript, but I had raised some concerns. All my previous concerns have now been addressed by the authors, and I compliment them on the outlier analysis and sensitivity analysis, which clearly involved a tremendous amount of work and constitute a contribution in their own right.

Given the novelty of SWOT data, the extent of the study, and the depth of the analyses performed, I have no reservations in recommending this work for publication in Nature. I look forward to seeing the paper in print.

Paola Passalacqua

(Remarks on code availability)

Referee #2

(Remarks to the Author)

Thank you for your detailed response. Overall, this study provides the first assessment of SWOT's performance in monitoring global river storage changes, which is remarkable. I am glad to see the uncertainties quantification and I think more in-situ data validations and uncertainty estimations are required in the future. It is also essential to account for the impacts of factors such as water surface slope, ice cover, and artificial structures. At the moment, I think it is difficult to fully rely on the results at a global scale. For this study, I only have some suggestions.

Extended Data Table 1. The analysis retains only observations with $xover_cal_q = 0$, substantially reducing the number of observations. As this flag reflects the application of crossover calibration rather than data quality, the rationale for excluding crossover-calibrated observations, and whether this filtering leaves some reaches with insufficient valid samples for reliable analysis, should be clarified. Is it possible for these unreliable crossover calibrations to become as robust as those at sea in the future?

Extended Data Figure 2. Regarding the authors' reply on the EIV regression, it would be helpful to clarify the first failure

case, which is characterized by very large width variations but minimal height changes. This pattern may correspond to typical floodplain dynamics, where river width can be highly sensitive to small stage variations while remaining physically meaningful. Could the rectangle approximation in such cases contribute to the observed underestimation of RSA in the Amazon and other regions?

Extended Data Figure 7. While the authors acknowledge SWOT's observational limitations, the magnitude of the deviations in some basin-scale ratios appears exceptionally large, raising doubts as to whether these factors alone can adequately explain the extreme values shown in Extended Data Fig. 7. Do basins with anomalous ratios systematically correspond to floodplain- or wetland-dominated regions, areas affected by seasonal ice cover, or known river width classification issues in the RiverSP C product, or could these discrepancies partly reflect unresolved issues in the authors' subsequent SWOT data processing methods?

(Remarks on code availability)

Referee #3

(Remarks to the Author)

I would like to thank the authors for their thorough and constructive responses to our previous comments. I have carefully reviewed the revised manuscript and the detailed rebuttal letter. The authors have successfully addressed my major concerns, and the quality of the manuscript has been significantly improved. The data presentation is now more robust, and the logic is clearer.

Overall, I believe this work makes a valuable contribution to the field and is suitable for publication. I am pleased to recommend the acceptance of this manuscript. However, before the final publication, I have a few minor suggestions and hope the authors can consider these minor points to further enhance the readability of the paper.

(1) In the Abstract (e.g., around Line 19), the authors should mention the scope of the river objects studied. To highlight the significance of this work to a broad audience, I suggest explicitly stating the proportion of global rivers covered by this study or using a phrase like "near-global scale" to define the spatial coverage more precisely.

(2) Line 47: the literature (<https://doi.org/10.1016/j.ejrh.2022.101020>) could be a strong support here.

(3) In the current version, Section 1 and Section 2 of the Results (specifically Lines 77–94 and Lines 121–135) contain a considerable amount of technical detail regarding data processing and methodological steps. While accurate, these details somewhat interrupt the flow of the scientific findings.

I recommend simplifying the description of data handling in the main text and moving the detailed methodological procedures to the Methods section. The Results section should focus more on the interpretation and implications of the findings (e.g., the patterns of sediment flux changes) rather than the mechanics of the calculation.

(4) The study references the reservoir representation from the SWOT Prior Lake Database (PLD), which is derived from the UCLA 2015-Lake product (<https://doi.org/10.1016/j.rse.2015.12.041>). It should be noted that this dataset primarily represents the state of reservoirs around 2015. Given that numerous reservoirs may have been constructed over the past decade, their water storage changes are captured by SWOT as river reaches. It might introduce non-negligible uncertainties into the estimation of river water volume changes at the local basin scale. I recommend adding a brief explanation or discussion in the Methods section (or another appropriate location) to acknowledge this potential limitation regarding the dataset's vintage and the potential impact of recently constructed reservoirs.

(5) Line 181 (figure caption): "Global observed SWOT..." should be "Global SWOT-observed..."

Chunqiao SONG (co-reviewed with Kai LIU)
NIGLAS, Chinese Academy of Sciences

(Remarks on code availability)

Referee #4

(Remarks to the Author)

I co-reviewed this manuscript with one of the reviewers who provided the listed reports.

(Remarks on code availability)

**Referees' comments:**

**Referee #1 (Remarks to the Author):**

This paper leverages new data from the SWOT mission over a water year to quantify variations
in water storage in rivers at a monthly scale. The work also cleverly utilizes these data to detect
the shape of river banks and thus the variations in cross section in the active portion of rivers as
water level varies. The authors also compare their results with previous storage estimates
obtained from modeling.

Exciting analysis and paper. This is the type of study I was really looking forward to seeing with
SWOT data —the opportunities these data offer for analyzing the global hydrologic cycle are
numerous, and this is an exciting first glimpse at what a full water year from SWOT looks like.
I very much like the idea behind this paper, as well as the approach and the metrics used to
quantify variations in storage in the world's rivers and the comparison to the modeling study. I
also found the concept of tracking water levels from low to high and thus the shape of the active
channel particularly exciting. Combined with water extent, water level measurements allow for
measurements of river shape from the lowest to the highest water surface elevation detected by
SWOT. Super cool.

However, I do have a major set of questions related to the SWOT centerlines and the potential
impact of their inaccuracies on the study presented. These questions are outlined below:

**Response to Referee 1:**

Dear Prof. Paola Passalacqua,

Thank you very much for your encouragements and this detailed feedback. All your comments
have been responded to in full and resulted in modifications and additions to our initial manuscript,
as described below.

In particular, while we did not specifically filter SWOT data based on centerline quality, we now
acknowledge that inaccuracies occur in deltas and braided systems, which may occasionally
affect river storage anomaly estimates. To address this, and motivated by one of your specific
comments, we performed a global outlier analysis of our dataset in Supplementary Information
and added caveats in the main text, highlighting reaches with reduced reliability. Figures S6–S8
now provide zoom-ins on representative challenging systems where centerlines may be off.

We very much appreciate the opportunity to improve and resubmit a revised version of our
manuscript, which we believe is now much stronger.

Sincerely,

Arnaud Cerbelaud, Jeffrey Wade, Cédric David, Tamlin Pavelsky, and Michael Durand, on behalf
of all authors

- Did you perform any selection of the reaches, or were all SWORD reaches included? From
what is written in the paper, it doesn't seem like a selection was made. I understand there was
some filtering based on flags on the water surface elevation, but it's unclear whether any checks
on centerline quality were performed. I ask because I've used SWORD extensively and have
retraced centerlines in many global river deltas, where SWORD centerlines were often quite
inaccurate. I've also examined centerlines in other rivers I work on and, in some cases,
observed centerlines placed in the floodplain (e.g., the meandering Trinity River has several
such reaches).

This is a good point. The only pre-selection (before data quality filtering) that we performed is on
the type of reaches included: type 1 (river) and 5 (unreliable topology). They represent around
174,000 out of the ~240,000 reaches in SWORD v16. We removed reaches of types 2 (lake off
river), 3 (reservoir), 4 (dam/waterfall) and 6 (ghost reach) where our approach is not necessarily
relevant and to avoid issues with SWORD centerlines as much as possible.

Beyond this selection, we did not further filter out reaches based on centerline quality. SWORD
centerlines can indeed be inaccurate in river deltas and braided river systems overall. These
issues are currently being addressed in SWORD v17 and v18 (in preparation). The equivalent
widths obtained on such braided systems can therefore result in under/overestimation of
storage changes (but not necessarily, see response about width calculations on the next page),
as illustrated in our new **Figures S6, S7 and S8** that represent the Ganges delta, the Niger
inner delta and the eastern part of North America (from Mexico to Canada, including the Trinity
river).

However, centerlines are unlikely a major issue for most non-coastal rivers given large-sample
WSE accuracy assessments, and these issues are not likely to impact a large percentage of
global reaches. To highlight this, we performed a **new full statistical analysis of RSA outliers**
(inspired by one of your later comments) in the new **Supplementary Information** (see on the
next page in response to your last major comment). We therefore added a caveat in the main
text:

**I. 183-188: In addition to in situ validation, we conduct a global statistical analysis of the**
**SWOT-derived Δ RSA using a tailored outlier detection approach. Our findings highlight**
**reduced reliability of RSA estimates at 3.4% of global river reaches, accounting for 8.8%**
**of Δ RSA but impacting average seasonality only by a 1.7-day shift. These outliers are**
**mostly located in complex and dynamic riverine environments (e.g., braided systems,**
**deltas), consistent with known limitations in the SWOT River Database (such as**
**centerline positioning quality; see section 5 of the Supplementary Information).**

We also added detailed limitations **I. 169-184 in Supplementary Information section 5.**

- Even if the centerlines were perfect—which they are not—some rivers are extremely dynamic.
The Ganges is mentioned frequently in the paper, and that system is highly dynamic, with
SWORD centerlines that are notably off.

You are right. The reduced reliability of our RSA estimates in dynamic riverine environments is
now acknowledged in the main text and the supplementary information as described above (**I.**
**183-188 in main text and in Supplementary Information section 5**).

- A related and more specific question concerns coastal areas: were delta reaches included, or

did the study stop upstream of deltas? If deltas were included, were variations due to tides
accounted for? If so, it might be worth analyzing those reaches separately from the rest of the
basin.

Delta reaches (that are not type 6) were included and the variations due to tide were not
specifically accounted for. For these few coastal reaches, we acknowledge that a refinement of
the hypsometric approach would be needed. We included a caveat in the Supplementary
Information:

**I. 60 in Supplementary Information: In addition, multi-channel structures are not**
**resolved, and river deltas are not specifically processed to account for their complexity**
**(e.g., anabranches) and the influence of ocean tides. We acknowledge that a refinement**
**of the hypsometric approach would be needed for coastal areas.**

I understand this is a global-scale study and that checking the accuracy of every reach is not
feasible. However, I would suggest double-checking especially those reaches that show
extreme (either large or small) variations in storage. The small variations, in particular, could be
cases in which the centerline is missing the channel altogether.

This is a critical point, as the quality of the centerline potentially affects all analyses performed
with the data. While the authors are likely aware of these issues, the current information in the
paper did not reassure me about how inaccuracies in SWOT might propagate through the
analysis and affect the storage and related metrics and how the authors may have accounted
for these issues in their analysis.

My suggestion is to include some evidence in the paper that centerlines were checked,
particularly for cases where extreme storage variations were observed. Additional figures with
insets showing the Ganges and other systems could help illustrate where the centerlines fall
relative to imagery and/or the SWOT water mask in raster form.

The SWOT widths are calculated from the total area of water identified in the pixel cloud data
around the prior river database (SWOT): in cases where no lakes near the river channel are
expected, the extreme distance is far from the channel centerline (10 times the river reach
width). The search rectangle extends half the river reach width in the cross-reach direction and
three node spacings in the along-reach direction from the node. A large extreme distance is
useful for wide river channels with sharp bends to ensure that all connected pixels in the river
channel are included in the pixel assignment (see **section 3.1.2.4 of the SWOT ATBD**
**document and its figure 7**).

Therefore, even when centerlines are off, water can still be detected even if relatively far away
from the centerline.

As you suggested, we performed a **new global statistical analysis of our storage variation**
**Δ RSA estimates** to detect mismatches between the SWOT network and our results. For this,
we developed and implemented a tailored outlier detection method based on the expected
increasing relationship between upstream drained area and the Δ RSA estimates we derive in a
basin. This analysis framework on extreme values (low or high) in our database and the
resulting limitations is now exposed in the **Supplementary Information (section 5)**.

We find that a great number of outliers are located in lower flow accumulation areas and close
to the outlets, indicating reaches located in river deltas (as suggested in your comment). We

note that the drained area originating from the MERIT-SWORD translation is also not always
fully reliable in these sections. Therefore, this additional statistical analysis highlights
mismatches or limited coherence between the reaches characteristics (topology, location of river
centerline which influences the computed widths variations) and the Δ RSA estimates derived
from the associated SWOT measurements, and overall the locations where our estimates may
not be reliable (see I. 183-188 in the main text, I.77-80 in **Supplementary Information**
**section 3** and **Supplementary Information section 5**).

Overall, we find that these less reliable estimates do not undermine the major findings of our
manuscript: they represent 3.4% of global river reaches, account for 8.8% of Δ RSA but impact
average seasonality only by a 1.7-day shift. In comparison, global uncertainties from SWOT
data and the hypsometric approach on Δ RSA can represent between 10 and 40% of the signal
(see responses to the other two referees on updated uncertainty quantification).

**Other comments/questions:**

Line 60: I understand that centerlines were traced for rivers wider than 30 m, but we don't trust
SWOT for rivers narrower than 50 m, correct?

We have processed all data that passed quality filtering based on current state of knowledge for
SWOT observations. Some of the final processed data available in our RSA product cover river
reaches narrower than 50 m that are present in the SWOT River Database. While the reliability
of SWOT observations over such narrow rivers remains to be fully demonstrated, recent work
presented at the SWOT Science Team Meeting (October 2025) suggests that SWOT can indeed
provide accurate measurements at times for rivers below 50 m in width.

Line 91: This is not exactly the river shape, but rather—as specified here—the river shape
between the lowest and highest water surface elevations. Unless these are ephemeral rivers,
this represents only a portion of the channel shape. I missed this earlier but effectively you
called it the active channel shape – it might be helpful to include this definition in **Extended**
**Figure 2 (at least in the caption) so that terminology is consistent.**

Thank you for this comment. We included the definition of active riverbed shape in the caption
of Extended Data Figure 2 as recommended:

Extended Data Figure 2. Conceptual diagram of SWOT-derived **active riverbed shape** and
river storage anomalies. a, W and WSE observations and assumed **active** channel bathymetry
for a given reach. \hat{A} corresponds to the unobserved cross-sectional area below the lowest
observed WSE.

Line 105: I understand these two systems may have comparable discharge, but one is covered
with permafrost and the other is not. I wonder if comparing systems within the same climatic
zone might be more meaningful.

This is a very fair comment, thank you. Indeed, they are very different river systems, including
different climatic zones. We also drew a parallel between the Orinoco and Congo rivers which
are in relatively similar climatic zones. The idea behind these comparisons is simply to expose
the breadth of active shapes observed by SWOT for rivers that have comparable discharge and

1 are often represented with rectangular cross-sections in hydraulic models. We conservatively
2 modified the main text accordingly:

3 L. 107-108: **Located in contrasting climates**, the Mississippi River showcases much smaller
spatial variability than the Yenisei River despite comparable average discharge and both being
highly managed.

Figure 1: I suggest changing the line style for the median—it's currently very hard to see, which
makes interpreting the figure difficult. For example, in the Ganges, the mean is outside the IQR
bounds, suggesting some extreme values are shifting the mean. Where did those come from?
And is the median underneath the mean?

Thank you for this helpful comment highlighting the difficulty in figure readability. We changed
the line style for the median active riverbed shape from thin solid to **large dashed** in Fig. 1.
When the mean is outside the IQR, like in the Ganges, it indeed highlights that the average
shape is biased by a few much larger values (that are outside the 75th quantile), which in the
case of the Ganges is due to downstream reaches in braided sections where the equivalent
widths are very large.

Figure 2: This is a very important figure, but it's hard to see anything at this scale beyond overall
color and variability (unless that is what you wanted to show). I suggest creating additional
insets that show major river basins in more detail so that centerlines can be seen and
analyzed—especially for Figure 2a.

Thank you for this comment. As part of our new outlier analysis of the SWOT-based RSA
database in Supplementary Information, we produced **Fig. S6-S8** to provide zoom-ins of Δ RSA
(as in Fig. 2b) on the Ganges delta, the Niger inner delta and the Mississippi delta.
This allows to check centerlines in areas where our approach seem to work well (e.g., the Trinity
river, where very few to no outliers are present in Fig. S8) and in other challenging areas
(braided systems) that are representative of issues in similar environments.

Apologies if I missed anything and I do hope these comments are helpful!

Paola Passalacqua

Thank you again for your constructive review.

**Referee #1 (Remarks on code availability):**

I did not install the code and only had a look at the github page to see the associated
documentation, which is comprehensive.

**Referee #2 (Remarks to the Author):**

The SWOT satellite is designed to observe Earth's surface waters and (sub-)mesoscale ocean
circulation, employing a novel Ka-band Radar Interferometer for the first time. This paper
focuses on mapping riverbank shapes and water storage changes of global rivers using SWOT,
which are also the main goals of SWOT. It is interesting to investigate and reveal the river
changes and their characteristics in different basin scales and the global scale. Although this
study provides the first near-global measurements of active river corridors and river storage
variability, I have some suggestions and concerns about SWOT and the results.

**Response to Referee 2:**

Dear Reviewer,

Thank you very much for this detailed feedback. All your comments have been responded to in
full and resulted in modifications and additions to our initial manuscript, as described below.

In particular, we re-ran our full study with updated uncertainty quantification based on a previously
published and verified methodology. The corresponding information and results have been
incorporated into the main text and the methods (with updated numbers and figures). We also
tamed the language and further discussed the limitations of the current SWOT data processing
and hypsometric approach.

Finally, we wish to highlight that even though an extensive analysis of the accuracy of SWOT river
products cannot be included in the present paper, comprehensive validation papers are currently
in review, and we would be happy to share the results with you if it would be helpful.

We very much appreciate the opportunity to improve and resubmit a revised version of our
manuscript, which we believe is now much stronger.

Sincerely,

Arnaud Cerbelaud, Jeffrey Wade, Cédric David, Tamlin Pavelsky, and Michael Durand, on
behalf of all authors

**Concerns about the goal of SWOT:**

Lines 427-438. After a series of filtering, 126,674 river reaches are retained from ~240,000
global river reaches. **More details should be given on how many reaches are filtered at each**
**step, and the reasons for their exclusion based on different quality criteria or conditions.** It would
also be valuable to analyze the extent to which SWOT has achieved its intended goals for
global river observations.

We added an **Extended Data Table 1** providing the number of unique reaches and
observations left out from the successive data quality filtering steps in the **Methods** part. Note
that many of the reaches that were removed are not rivers per se, but are lakes or reservoirs
that form part of the river network or are so-called ghost reaches, extremely short reaches at the
beginnings and ends of rivers which are used for data processing purposes but are not intended
to contain data. Thus, the number of “missing” river reaches in our analysis is substantially
smaller than the accounting above suggests.

Analyzing the extent to which SWOT has achieved its intended goals is the responsibility of the
SWOT “Science Team” in support of space agencies and goes far beyond what a small group of
co-authors can accomplish. However, it is our hope that this paper will contribute to the
assessment for SWOT’s ability to track river water storage changes. We clarified this in the text:

**I. 190: While assessing compliance with SWOT mission requirements is beyond our**
**scope, we hope these results will contribute to advancing the mission’s scientific**
**objectives.**

Lines 480-482. The scientific requirements for river width and WSE from SWOT are 30 m and
0.1 m, respectively. According to the Science Requirements Document, one of SWOT’s
requirements is to map rivers wider than 100 m, with an extended goal of 50 m. **Have the**
**requirements been validated with other technologies and measurements? Could the**
**uncertainties of these two variables be used to provide uncertainty in storage changes?** In the
input data provided by the authors, I examined three basins (Amazon, Yangtze, and Ganges)
and found that the typical median WSE uncertainty (*wse_u*) is about 0.4~0.6 m, and ~0.17 m for
rivers 50~100 m and >100 m wide. **Do these results indicate that SWOT has actually met its**
**stated requirements and goals? If not, does this imply that the estimates for smaller rivers may**
**be inaccurate?**

As you suggested, we provided **new full uncertainties in storage changes RSA from**
**uncertainties in WSE and width.** For this, we **re-ran our full study with updated SWOT**
**uncertainty quantification based on a previously published and verified methodology**
(Coss et al., 2023) incorporated in FLaPE-Byrd. The corresponding information and results have
been incorporated into the main text and the methods (with updated numbers and figures). This
was one of the most time-consuming aspect of this revision but adds definite robustness to our
study.

The validation of SWOT’s performance and mission requirements with other technologies and
measurements is an ongoing investigation undertaken by JPL/CNES and several groups
worldwide and represents a major undertaking. The responsibility of the SWOT “Science Team”
is to analyze the extent to which SWOT has achieved its intended scientific goals, and we
therefore analyze the data and produce river water storage change estimates based on the

current best knowledge, including the latest uncertainties, regardless of their magnitude relative
to requirements.

While **comprehensive validation papers are currently in review**, we would be happy to share
the results if it would be helpful. In essence, comparison with a suite of >1000 gauges suggests
that SWOT likely meets its science requirements for water surface elevation. However, we
cannot include that extensive analysis in the present paper and assessing the validity of the
stated requirements is unfortunately beyond the scope of this study.

We referred to this in both the main text and the methods:

9 L. 114: These measurement characteristics are currently being investigated in the context of
10 large-scale in situ validation efforts⁴².

**I. 190: While assessing compliance with SWOT mission requirements is beyond our**
**scope, we hope these results will contribute to advancing the mission’s scientific**
**objectives.**

14 L. 548: The simultaneous measurements of river width and water surface elevation provided by
15 SWOT represent an unprecedented advancement in global hydrology, but their validation
remains an ongoing and highly complex effort due to the mission’s global scope and novel data
products⁴².

We wish to add that some of the final processed data available in our RSA product cover river
reaches narrower than 50 m. While the reliability of SWOT observations over smaller rivers
remains to be fully demonstrated, recent work presented at the SWOT Science Team Meeting
(October 2025) indicates that SWOT can indeed provide accurate WSE measurements at times
for rivers below 50 m in width.

Line 733, Extended Data Figure 1. Are all the rivers without measurements from November to
May located in the Arctic? **How about rivers in complex mountainous terrains and high-altitude**
**cold regions, such as the Tibetan Plateau**? Can you give the distribution or analysis of the rivers
that have more missing data?

Good question, which motivated clarifications in the caption of this figure. They are not all
located in the Arctic (Arctic basins are codes 25 to 27 in Europe, 31 to 35 in Siberia, 71, 72 and
81 to 91 in North America/Greenland). Most parts of basins 36 and 47, 48 and 49 are also
without measurements from November to May (see Extended Data Figure 1) and are located in
Mongolia, North-Western China, Tibet, the Himalayas, i.e., complex mountainous terrains and
high-altitude cold regions. We added a mention to this in the Extended Data Figure 1 caption:
Extended Data Figure 1. [...]. **Regions of Mongolia, North-Western China, Tibet, the**
**Himalayas, i.e., complex mountainous terrains and high-altitude cold regions, also lack**
**post-filtering data typically from November to May.**

Extended Data Figure 1 precisely gives the distribution of river basins that have missing data,
and the cadence of acquisition where data passed quality filtering. Users are encouraged to
download our .csv/.shp database and analyze which rivers lack data post filtering for more
details.

**Concerns about the processing and analysis of river activities:**

Line 100, Figure 1. The interquartile ranges of global rivers vary significantly. For example, for
Congo River, it could be larger than 1000m, while for the highly active Amazon River, it could be
less than 1000m. **Is it affected by the SWOT's capabilities and uncertainties?**

It could be, but to a very small extent. This is mostly the reflection of the geomorphology of each
river along its course. Both the Amazon and the Congo are very large downstream, but the
Amazon is consistently large (almost always above 1500-2000 m in width) because it starts off
large from the Ucayali and the Marañon rivers, while the Congo starts off from the Lwalaba river
which is comparatively smaller, and thus narrower upstream (1000 m). We added a note in the
main text:

**L. 108: Varying interquartile ranges indicate contrasting geomorphologies along river**
**courses (e.g., Amazon versus Congo basins).**

Line 113. **How is frozen water defined**, and how to judge whether the observations are over
frozen water or not? It is useful to know whether the river observations can be used and are
reliable.

Frozen water is defined with the *ice_clim_f* > 0 flag:

I. 436: Observations where **external** climate indicators imply the presence of ice or snow
(*ice_clim_f* > 0) are also filtered.

This corresponds to a climatological ice cover flag indicating whether the reach is likely to be
ice-covered on the day of the observation based on external climatological information (see
<https://doi.org/10.1038/s41586-019-1848-1> for more information), not the SWOT measurement
itself.

We suggest the reader refer to the SWOT product description document for detailed information
on the quality flags ([https://archive.podaac.earthdata.nasa.gov/podaac-ops-cumulus-docs/web-](https://archive.podaac.earthdata.nasa.gov/podaac-ops-cumulus-docs/web-misc/swot_mission_docs/pdd/D-56413_SWOT_Product_Description_L2_HR_RiverSP_20250224a_RevC_clean_sig_final.pdf)
[misc/swot_mission_docs/pdd/D-](https://archive.podaac.earthdata.nasa.gov/podaac-ops-cumulus-docs/web-misc/swot_mission_docs/pdd/D-56413_SWOT_Product_Description_L2_HR_RiverSP_20250224a_RevC_clean_sig_final.pdf)
[56413_SWOT_Product_Description_L2_HR_RiverSP_20250224a_RevC_clean_sig_final.pdf](https://archive.podaac.earthdata.nasa.gov/podaac-ops-cumulus-docs/web-misc/swot_mission_docs/pdd/D-56413_SWOT_Product_Description_L2_HR_RiverSP_20250224a_RevC_clean_sig_final.pdf)).

Line 119. Could the authors clarify why δA is computed relative to the median WSE, and how
the choice of zero reference might affect the RSA results?

This is a purely arbitrary choice that does not affect the results in any way. A reference is
needed to provide storage changes (we do not measure absolute storage). We could have
chosen the first measurement in the series. An addition was made in the methods for added
clarity:

**I. 544: The choice of the median as reference to present the changes in RSA does not**
**affect the reach-, basin- or global-scale seasonal analysis of RSA nor the estimation of**
**river storage variability ΔRSA .**

Lines 125-127. Which method is used for the interpolation of missing SWOT observations?
Since the dates of overpass are irregular and the missing observations vary with the latitude,
especially for the rivers over the Arctic area, can the method be applied in these conditions?

This is detailed in the **Methods I. 533 to 539**, and include a mention that irregular overpass times
are accounted for. We initially chose to interpolate linearly everywhere and every time, and then
realized it wasn't in agreement with how river storage evolves during the winter season in rivers
that freeze. Therefore, we chose to interpolate with a forward fill whenever and wherever the ice

1 flag is on, keeping a stable RSA throughout winter (which doesn't mean that the river stops
2 flowing):

3 L. 535: This ensures that the resulting RSA at each reach is reflective of the observational
cadence of SWOT. A forward filling approach is adopted when water is "likely fully ice covered"
(ice_clim_f = 2), to maintain a stable RSA during the winter months (e.g., from November to
May in the Arctic), and a linear interpolation between the two surrounding valid observations is
performed otherwise.

Line 132. SWOT provides one valid observation every 28 ± 4.6 days on average, but it varies
with the latitude, and the revisit rate is about 10 days. How did you get the monthly variability?
Since we eventually interpolate the RSA estimates to the true observational cadence of SWOT
(see above and in the Methods), all reaches have at least 1 to 2 data points per 21-day cycle,
and therefore at least 1 to 3 data points per month. The monthly variability is then calculated as
a monthly mean.

The monthly RSA therefore originates from either "true" data points, a mix of "true" and
interpolated data points, or interpolated data points only (such as in the winter months for frozen
rivers). This is described in the main text:

I. 130-135: Occasional missing SWOT observations that were removed by data quality filtering
were subsequently infilled by interpolating the observed volume change time series to SWOT
overpass dates, accounting for its irregular temporal sampling (between 3 and 21 days
depending on the latitude). Finally, we recompute the volume changes relative to the
interpolated time series average $\bar{\delta V}$ at each reach to produce zero-mean river storage
anomalies (RSA) at SWOT overpass dates t_{SWOT} (Eq. 1; Extended Data Fig. 2d), and for each
24 month m [...]

Lines 136-140. Since there is only one year of observation of SWOT, and for each month, there
may be only a few observations, is it enough to analyze the maximum river storage for each
28 month? Some extreme river changes may be missed. And the seasonal patterns do not always
follow the latitudinal gradients, especially for rivers in the Arctic and the Amur basin. Except for
the Amazon River and Congo River basins, **how many basins show patterns consistent with**
**current understanding and how many are inconsistent?**

This is a fair point and it is indeed a challenge in some areas, as the monthly RSA estimates at
certain times of the year can originate from a single true SWOT data point as well as limited
interpolated data points, or even from interpolated data points alone.

We added several caveats in the main text:

I. 115: In addition, one year of observations at an average repeat of 28 days (after data filtering,
Extended Data Fig. 1) likely obscures occasional high extremes [...]

I. 177-178: **Overall, SWOT's limited effective temporal sampling can cause the monthly**
**and annual RSA ranges to miss part of the true variability.**

I. 214: Temporal correlation and underestimated magnitude against the prior low volume
scenario³ are both also apparent regionally, **and inconsistent patterns and/or ranges were**
**found in several Arctic basins, Central Asia, southern South America, the Nile basin, and**
**the Western U.S. (Extended Data Fig. 6, 7, 9 and 10).**

The inconsistent basins are now detailed in Extended Data Figures 3, 6, 7, 9 and 10: several

Arctic basins, Central Asia, the south of South America and the Western U.S. mostly, hence
helping shed light on expected opportunities for improvement beyond the first year of SWOT
data alone.

Line 168, Extended Data Figures 3. Low correlations also exist in the Rio Parana and Colorado
River basins.

Yes indeed. We completed the relevant sentence in the main text:

I. 173: Seasonal validation confirms that SWOT RSA appears reliable across most of the globe,
except for parts of the Arctic, **southern South America and the Western U.S.** (Extended Data
Fig. 3).

This was already present in the Methods:

I. 575: Discrepancies in basins such as the Paraná or the Colorado could indicate limitations in
SWOT's current ability to consistently delineate water extent in complex and heavily managed
hydrological settings.

Line 461. Formula (1) in the Methods section is theoretically applicable, but for long river
reaches or rivers with variable slopes, it may introduce errors. **Could the authors clarify how
river slope is accounted for in these cases, given that SWOT can, in principle, measure slope?**

Very good question. Longitudinal slope doesn't affect the geometric definition of cross-sectional
area A (formula or Eq 1), but because we use reach-averaged height and width over a long (10
21 km) reach, longitudinal variation (e.g., at the node scale) in stage and width can make Eq (1) a
22 biased estimate of the true mean cross-sectional area.

We did acknowledge this limitation already in the Methods, but now added more information
earlier on in the same section:

I. 57 in Supplementary Information: The hypsometric approach used in this study provides a
simplified product as it leverages averaged width and WSE over 10-km long river reaches,
removing the intra-reach spatial variability. The method also inherently assumes a symmetrical
profile which only provides partial information on the true shape of the banks and extent of the
floodplains.

**I. 478-480: We note that Eq. (1) is only an approximation since measurements are**
**averaged at the reach scale, overlooking finer-scale longitudinal variations in width and**
**height.**

Note, however, that slope would only impact the volume calculation if one was using one or a
few nodes. As the nodes span the whole reach, storage is computed directly without any
assumption about slope. Even if one were to argue that longitudinal slope should enter into the
computation of volume changes $\delta A * L$, the reach length L should be adjusted with the cosine of
the river slope s (volume of a parallelepiped), s being the tangent. As median/average river
slopes s are 0.06%/0.26% worldwide, this would mean a $\frac{1}{\sqrt{1+s^2}} < 10^{-6}$ adjustment on volumes.
Even with a slope of 4 to 6% on very steep rivers, this would only be a 10^{-3} adjustment, well
below the measurement and hypsometric uncertainties. This could thus be neglected.

Lines 491-496. In practice, **how do the authors know the width measurements are unreliable,**
**leading to the initial EIV regression failing?** Or is it that the initial EIV regression fails, indicating

unreliable width measurements? More details should be verified, and the distributions and
circumstances of the 3,850 reaches can be concluded.

In ReachObservations_jw.py available in the github repository, we resort to rectangular fits when
one of the slope terms of the regression in one of the subregions is (i) greater than 10,000 or (ii)
negative; which means (i) when width increases by 10 km when height increases only by 1 m
(very flat region), or (ii) when width decreases when height increases (unphysical). For these
reaches, we conclude that width variations are too extreme (too small or too large) or noisy and
that only height variations are valid (which is consistent with the findings of the CNES/JPL
processing team that was presented at the SWOT Science Team meeting in October 2025:
heights are much more reliable than widths). Because we do not want to discard these reaches,
we keep the median width and resort to a rectangular fit, keeping only the height variations.
We completed the following sentence:

I. 508: In some circumstances, the initial errors-in-variable regression fails to converge or
produces implausible results (**m_i terms either greater than 10,000 or negative**), [...]

Line 639. Some basins in Extended Data Fig. 7 exhibit ratios that are unusually large or small,
whereas theoretically these ratios should mostly range near 1 (e.g., 0.75–1.5). **Could the**
**authors explain the cause of these deviations?**

The many potential causes of the deviations between SWOT and MeanDRS are extensively
described in the main text and methods. However, we added a reference to the WMO 2024
State of Global Water Resources report that provides information on regional hydrologic
conditions in 2024 and reveals a strong agreement with our findings (i.e., SWOT’s regional
underestimations):

24 L. 218: Exceptional drought conditions in the Amazon basin since 2023⁵² **very** likely impacted
the observed global Δ RSA (Extended Data Fig. 6) as several major rivers fell below historical
records^{16,53}, while conditions continued to worsen in 2024⁵⁴. **In addition, the Lena and Yenisei**
**basins account for most of the discrepancies between SWOT and the model from**
**November to March (Extended Data Fig. 6). SWOT-based variability for 2023-2024 is likely**
**rationaly lower than that of past climatological averages. In fact, the world’s river conditions**
**in 2024 were reported “much below” average in South Africa, the Congo, the Nile, the**
**Mackenzie, Southwestern North America, the Amazon, and the Lena, in strong agreement**
**with SWOT’s underestimations against the prior model (Extended Data Fig. 7 versus**
**figure 7 of the 2024 WMO State of Global Water Resources Report⁵⁵).** A longer data record
from SWOT’s planned 3-year duration will help refine the elevation–width approach and
measurement uncertainties, **and allow the analysis of potentially more representative**
**hydrological years.** The inconsistency observed between SWOT-derived and modeled RSA
may also originate, in part, from contrasted definitions of river storage between SWOT’s
observational perimeter and the model structure (see **Supplementary Information**).

39 L. 68 in Supplementary Information: SWOT’s observational scope remains challenging to fully
characterize, particularly in complex hydrological environments. The discrepancies observed
between SWOT-derived and modeled RSA may originate, in part, from contrasted definitions of
river storage between SWOT’s observational perimeter and what models incorporate. While
models may systematically integrate all water flowing through the river–floodplain continuum,
SWOT’s retrievals depend on surface water detection and **altimetric** retracking algorithms that

may exclude or inconsistently capture key dynamic zones (particularly in the Version C RiverSP
products). In areas like the Arctic wetlands and along snowy riverbanks in transitional periods,
or in South America's Pantanal during the rainy season, SWOT's classification of river extent
can vary significantly. More detailed and large-scale evaluation of SWOT river width detection
will likely clarify how comparable these datasets truly are.

6 L.658: With more SWOT observations spanning full annual cycles (when version D products are
7 available) and more local validation endeavors, clearer explanations will emerge.

Our findings reflect the current state of knowledge in measurements with the novel KaRIn radar
interferometer. With more SWOT data in the next years, we will most likely be able to better
disentangle the reasons for these differences.

Line 693. In the last part of the Methods, does the term "retracking algorithms" correspond to
the same concept as retracking in nadir altimetry?

It does refer to the calculation of the height above the geoid from altimetry. But in this context, it
refers to the SWOT processing of elevation, which is different from its nadir counterpart. This
was edited for clarity (now in the SI):

17 L. 71-73 in Supplementary Information: While models may systematically integrate all water
flowing through the river–floodplain continuum, SWOT's retrievals depend on surface water
detection and **altimetric** retracking algorithms that may exclude or inconsistently capture key
dynamic zones.

**Concerns about the comparison with MeanDRS:**

The authors show that the storage changes of global rivers estimated by SWOT are always
lower than the lowest estimates of MeanDRS. They attributed the discrepancy to droughts in the
Amazon basin. However, this explanation is insufficient to account for the underestimation
observed in many other basins. In addition, considering Amazon's dominant contribution, can
the riverbanks and floodplains really be distinguished in dense rainforest regions? **Could such
potential misidentification lead to severely inaccurate storage change estimates?**

Most storage changes of global rivers estimated by SWOT are lower than the lowest estimates
of MeanDRS (Ext data fig. 7). We attributed the global discrepancy mostly to the Amazon
drought, considering the basin's immense contribution to the global Δ RSA. We wish to clarify
that we do not claim in the manuscript that this justifies other basins' underestimations.
For the Amazon, indeed, the riverbanks and floodplains are not necessarily consistently
distinguished everywhere by SWOT. Hotspots of the Amazon River water storage changes are
in large floodplains that the SWOT RiverSP Version C product most likely does not fully capture.
We acknowledge that our goal is not to capture all the storage changes in the basin. We only
wish to provide what SWOT saw, and to compare it with what models simulate. This is fully
described now in the manuscript:

I. 111: However, **the shapes only reflect what SWOT observed during 2023-2024**. Notably,
floodplains appear to be rarely captured in the elevation-width relationships (Fig. 1). Results
also reflect early-stage limitations in both our three-domain piecewise approach and SWOT's
current measurement capabilities and uncertainties. These measurement characteristics are
currently being investigated in the context of large-scale in situ validation efforts⁴².

I. 229: The inconsistency observed between SWOT-derived and modeled RSA may also
originate, in part, from contrasted definitions of river storage between SWOT's observational
perimeter and the model structure (see **Supplementary Information**).
I. 71 in Supplementary Information: While models may systematically integrate all water flowing
through the river–floodplain continuum, SWOT's retrievals depend on surface water detection
and **altimetric** retracking algorithms that may exclude or inconsistently capture key dynamic
zones (particularly in the Version C RiverSP products). In areas like the Arctic wetlands and
along snowy riverbanks in transitional periods, or in South America's Pantanal during the rainy
season, SWOT's classification of river extent can vary significantly. More detailed and large-
scale evaluation of SWOT river width detection will likely clarify how comparable these datasets
truly are.

Since different basins may exhibit distinct behaviors, performing a global-scale comparison may
be too challenging to analyze. Moreover, the global analysis may be affected by the irregular
temporal sampling induced occasional missing observations. It would be more reasonable to
include an analysis focusing on one or a few representative basins, which could also avoid the
influence of seasonal ice cover.

Thank you for this suggestion. However, we do not wish to restrict the analysis to a narrower
range of basins, and the editor specifically asked us not to do so. The scope of our manuscript
is intentionally global, which means the study includes uncertainties and limitations inherent to
the various hydroclimates and river morphologies in the world. While we cannot necessarily
disentangle all the current uncertainties, these global results are part of a process to help refine
all of the SWOT products in the future.

Although the authors studied the correlation of water storage changes between SWOT and
gauges, I don't see the estimation of the uncertainties for SWOT- or MeanDRS-derived water
storage changes. As a result, it remains uncertain whether SWOT offers advantages compared
to current approaches.

You are right. As mentioned above, we added a **full uncertainty quantification of our RSA**
**estimates** in this revised version of the manuscript (in the text and the figures, with details in the
Methods). The global Δ RSA estimate is now $313 \pm 129 \text{ km}^3$ (uncertainties are also provided at
the reach and basin scales). MeanDRS standard deviation climatology is detailed in the text and
figures as well.

Whatever the uncertainties, SWOT offers the definite advantage of being observational, with
joint measurements of water extent and elevation using one single sensor, not a model, and
densely covering the planet.

Lines 198-199. I can't see this information from Figure 4b.

The less variable green slices in Fig. 4b (i.e., the simulated years with less volume variability)
are going from -200 to +200 km^3 roughly. This information is also available in the Zenodo repo
(file *MeanDRS_slice_global_summary.csv*). We did not wish to add labels for the years 1984-85
and 1985-86 on Fig. 4b, which would clutter the visual.

Lines 202-204. At the annual scale, the Δ RSA in the Amazon basin has the largest impact on

the observed global Δ RSA. However, at the monthly scale, what about the RSA in the Amazon
basin compared with the global RSA? For example, in Figure 4, which basins primarily account
for the discrepancies between SWOT observations and model estimates?

We completed Extended Data Figure 6 with **two new graphs (6a and 6b)**, now showing the
monthly RSA regional contributions for both SWOT and MeanDRS. The basins primarily
accounting for the discrepancies are the Amazon, the Lena and the Yenisei (especially from Nov
to March).

**I. 217: Regional contribution to monthly offset between SWOT and modeled RSA can be
visualized in Extended Data Fig. 6.**

**I. 220: In addition, the Lena and Yenisei basins account for most of the discrepancies
between SWOT and the model from November to March (Extended Data Fig. 6).**

Line 218, Figure 4. There is a larger discrepancy between SWOT RSA and those of the model
from November to February than in other months, and there is also a lot of missing data during
this period. Any correlation between them?

**Excellent point. Yes, there is a connection indeed. See response above about the Lena and the
Yenisei which explain most of the discrepancy (Extended Data Fig 6a and 6b).**

Most of the time, the hydrological model underestimates the RSA due to incomplete
understanding and insufficient runoff and other input data. However, compared with the model's
lowest estimate, the SWOT RSA is even lower. What does this mean?

**This study may provide an initial step towards answering your question. It could mean that
overly low celerity values (that are often used to attenuate runoff peaks in river modeling) are
obtained from model calibration, increasing residence time and inflating storage compared to
SWOT (cited I. 234-237). Maybe even the lowest volume scenario (i.e., with the highest celerity)
in the model still features celerities that are too low. This is just a conjecture and will need to be
disentangled with more annual cycles of SWOT.**

Line 650, Extended Data Figures 10. There appears to be a temporal lag between the SWOT
and MeanDRS RSA time series, with some regions showing a lag exceeding three months.
Could the authors clarify the underlying cause of this lag?

**This is a good point. The potential underlying causes of these large lags are exposed in the
main text (see responses above and new additions below) and in the methods:**

**I. 575: Discrepancies in basins such as the Paraná or the Colorado could indicate limitations in
SWOT's current ability to consistently delineate water extent in complex and heavily managed
hydrological settings.**

**I. 653: The Amazon River basin and the Yenisei and Lena River basins show some of the
largest differences between SWOT and MeanDRS, likely due to a record drought in the Amazon
and challenges in resolving partially to fully frozen arctic rivers from SWOT, respectively.**

**I.675: We note that lags between SWOT and MeanDRS RSA time series can also be the
result of monthly lumped routing in MeanDRS, where runoff is accumulated from
upstream to downstream without accounting for horizontal travel time from land to rivers
or within the river system.**

I. 74 in Supplementary Information: In areas like the Arctic wetlands and along snowy riverbanks
in transitional periods, or in South America's Pantanal during the rainy season, SWOT's
classification of river extent can vary significantly.

Extended Data Figures 7–10. Basins in the Mongolia region show results that differ from those
of the surrounding basins. Could the authors clarify the reason for these discrepancies?

Several reasons may come to play here. First, the model simulation in these areas did not
benefit from gauge correction and are highly uncertain (see Collins et al., 2024). Second, there
are very few rivers in SWOT in the Mongolia region (pfaf codes 36, 47), most of them being
narrower than 100 m, potentially ephemeral and located on a high plateau. There is thus a great
deal of uncertainty in both of the products being compared: therefore we are not comfortable
conjecturing here, and did not modify the text.

Overall, the first observed global river bank shape and storage anomaly is derived by the SWOT
observations from October 2023 to September 2024, and the river storage changes are
compared and verified with the hydrological model estimates. Both annual and monthly basin
scale RSAs are emphasized. Beyond the first glance, some new insights could be highlighted.
SWOT is capable of mapping rivers wider than 30m at a global scale. However, its temporal
coverage is limited, and measurements of narrow rivers (less than 100 m), frozen rivers, or
rivers in complex terrains may still be challenging. These limitations may contribute to
unexpected patterns in the global- and basin-scale analyses presented in this paper. SWOT has
already demonstrated its ability to capture finer-scale ocean dynamics. **Therefore, over rivers, is**
**it possible to reveal finer-scale spatial changes, according to different river channel**
**characteristics and surrounding circumstances**, especially in typical basins?

Contrary to the ocean community, SWOT hydrology processing is much more recent: there is
much work to be done on the prior river (and lake) database, the signal processing, and the
filtering algorithms. Whether it can reveal finer-scale spatial changes is still uncertain, but
progress is being made. With more observations across annual cycles, uncertainties may be
untangled in hope to obtain more reliable storage change estimates at fine scales. Therefore,
until these uncertainties are better constrained, we believe it is premature to assert any
definitive implications.

**Referee #2 (Remarks on code availability):**

Several steps (2, 5, 7, 10, and 12) in `tst_pub_repr_all_Wade_etal_2025.sh` failed to run, and
outputs using the provided SWOT data did not fully match output.zip (e.g., basin 11, Step 1:
`m_1 = 68.3843, fit_method = simple` vs `m_1 = 65.5352, fit_method = set`). More descriptions
about the data input and output could be suggested.

The last CI/CD test (V12) ran successfully in May 2025 (see github repo). Since then, when re-
running the entire study with added uncertainty in November 2025, we did also observe very
minor changes in the EIV fits. If the coefficients of some regressions (slope and intercept)
appear to change significantly, they mostly compensate and the effective changes on the shape
and resulting flow area calculated are very small: the 50th/90th global percentile change in δA
compared to the initial results is 0.10%/0.32%, so well below the uncertainty range.

We found that these occasional changes were caused by changes in NumPy and SciPy
versions (1.26.4 for NumPy at least is needed, 1.12.0 for SciPy). Indeed, Numpy changed the
implementation of *var*, *std*, and *cov*, while SciPy's *curve_fit* optimization algorithm was modified.
Since the FLaPE-Byrd code (i) performs nonlinear optimization (*scipy.optimize.minimize*,
*curve_fit*), (ii) uses linear algebra and covariance (*numpy.var*, *numpy.cov*), (iii) relies on floating-
point edge cases (division by mXY, near-zero gradients) and has branching logic that switches
methods based on thresholds (>10,000, <0, etc.), and (iv) uses iterative solvers in SciPy that
changed, it can absolutely change occasional final outputs.
We are currently updating the Zenodo repo and project Github for full transparency and
replicability.

**Referee #3 (Remarks to the Author):**

Review of Manuscript Nature-2025-07-19902: "Wide-Swath Altimetry Maps Bank Shapes and
Storage Changes in Global Rivers"

This manuscript presents an analysis of the intra-annual variability in global river water storage
using the SWOT satellite observations, providing a straightforward view of river corridor
morphology and seasonal storage changes at a global scale. The work uses a critical dataset to
address a long-standing challenge in hydrology, and its findings have significant implications for
hydrological modeling, water resource management, and climate change studies. The research
on global-scale riverbank morphology has the potential to offering a novel perspective of
understanding fluvial geomorphology. However, despite its groundbreaking nature, the
manuscript in its current form has several fundamental weaknesses that affect the credibility of
its core conclusions.

**Response to Referee 3:**

Dear Reviewer,

Thank you very much for this detailed feedback. All your comments have been responded to in
full and resulted in modifications and additions to our initial manuscript, as described below.

In particular, we re-ran our full study with updated uncertainty quantification based on a previously
published and verified methodology. The corresponding information and results have been
incorporated into the main text and the methods (with updated numbers and figures). In addition,
we re-ran our study in ten contrasted and representative basins under three additional degraded
input uncertainty scenarios. These additional sensitivity tests reflect potential SWOT degraded
measurement conditions. This analysis was added in Supplementary Information. We also tamed
the language and further discussed the limitations of the current SWOT data processing and
hypsometric approach.

Finally, we wish to highlight that even though an extensive analysis of the accuracy of SWOT river
products cannot be included in the present paper, comprehensive validation papers are currently
in review, and we would be happy to share the results with you if it would be helpful.

We very much appreciate the opportunity to improve and resubmit a revised version of our
manuscript, which we believe is now much stronger.

Sincerely,

Arnaud Cerbelaud, Jeffrey Wade, Cédric David, Tamlin Pavelsky, and Michael Durand, on
behalf of all authors

**Major comments:**

1---A core strength of this research is SWOT's near-global coverage, particularly its novel
capability to map riverbank morphology. However, the manuscript does not fully leverage this
advantage. The spatial analysis lacks the necessary depth and detail to investigate the
heterogeneity of river characteristics across different climate zones (e.g., arid vs. humid,
mountainous vs. lowland) and within different sections of river basins (upstream, midstream,
downstream). For instance, Figure 1 presents a highly generalized overview by averaging
riverbank shapes along the entire mainstream of major rivers, which masks significant spatial
variability. The authors should delve deeper into the data to uncover new geographical patterns
or hydrological process insights that are only possible with SWOT's high-resolution
observations. For example, an analysis of typical bank morphologies in different geomorphic
units or climatic settings and their impact on water storage capacity would more effectively
demonstrate the value of this new observational tool.

Thank you for this very interesting comment. We agree that SWOT's high-resolution
observations are very promising, and that a deeper analysis of bank morphology in relation to
geomorphic characteristics and climates would be highly valuable to uncover new patterns.

However, the goal of this study is to advance SWOT's scientific objectives by deriving river
water storage changes, a long-anticipated but not yet demonstrated capability. We therefore
chose to focus on this variable and to compare SWOT-based estimates with the only publicly
available model-based estimates, with riverbank morphology serving primarily as a means to
that end. As a result, uncovering new geographical patterns or hydrological processes lies,
unfortunately, far beyond the scope of this paper. Surely the reviewer will appreciate that a
comparison with a state of knowledge on global river storage variability is at least equally
interesting and necessary.

We are working on an additional manuscript that specifically delves into this other topic—
indeed, the complexity of analysis required to do it justice would be impossible in the current
manuscript.

2---The study's reliance on a single hydrological year (2023–2024) to quantify global river
storage change is its most critical weakness. An anomalous year, heavily influenced by a
historic drought in the Amazon, cannot represent the long-term mean or range of variability for
global rivers. This makes it difficult for readers to determine whether the significant discrepancy
between SWOT observations and model estimates is due to an underestimation by SWOT, an
overestimation by previous models, or simply the effect of this single extreme event. The
authors should conduct a more nuanced regional analysis. For the study period, were other
major basins (e.g., the Congo, Mississippi) experiencing normal or even flood years? If so, how
do SWOT's estimates in those basins compare to models or gauge data? Such regional
comparative analysis would help to disentangle the impact of specific extreme events from
systematic observational or model biases and provide a more objective assessment of SWOT's
capabilities.

We entirely agree, and we stressed all your relevant criticisms about having only one year of
data in the main text and methods, including new additions:

I. 115: In addition, one year of observations at an average repeat of 28 days (after data filtering,
Extended Data Fig. 1) likely obscures occasional high extremes, [...].

I. 238: **SWOT-based river storage change estimates still face numerous uncertainties**, but
the mission offers a new path forward: **with more observations across annual cycles**,
uncertainties may be untangled in hope to build more physically grounded, globally consistent
models of river dynamics.

7 L.630: Regional hydroclimatic variability (e.g., El Niño periods) exerts a substantial influence on
comparisons between a single year of SWOT observations and long-term gauge-corrected
simulated means, as the study period captured by SWOT (2023-2024) may represent conditions
that are significantly different from climatological averages.

Information on regional hydrologic conditions in 2024 from external data have been provided by
the WMO in their recent 2024 State of Global Water Resources report (released in Sept. 2025).
Therefore, following your suggestion, we added in the main text a reference to figure 7 of that
report, which reveals a strong agreement with our findings (i.e., SWOT's regional
underestimations):

**I. 223: In fact, the world's river conditions in 2024 were reported "much below" average in**
**South Africa, the Congo, the Nile, the Mackenzie, Southwestern North America, the**
**Amazon, and the Lena, in strong agreement with SWOT's underestimations against the**
**prior model (Extended Data Fig. 7 versus figure 7 of the 2024 WMO State of Global Water**
**Resources Report⁵⁵). A longer data record from SWOT's planned 3-year duration will help**
**refine the elevation–width approach and measurement uncertainties, and allow the analysis of**
**potentially more representative hydrological years.**

Thank you very much for this helpful comment.

3---The study is based on the SWOT River Database (SWORD), but global river systems are
heavily regulated by reservoirs, which have a massive impact on river storage variability. The
manuscript fails to clarify how reservoirs are handled in the analysis. **Were reaches**
**corresponding to reservoirs entirely excluded**, or were they treated as natural river segments?
Did the authors use an existing global reservoir inventory (e.g., GDW, GeoDAR, GOODD) to
precisely identify and separate natural reaches from regulated ones? A clear quantification and
distinction of this regulated water volume is essential for an accurate estimation of natural global
river storage changes, and this must be detailed in the Methods section.

**Reaches corresponding to both reservoirs and natural lakes that are part of the river system**
**were deliberately excluded. This is fully referenced in both the main text and the methods:**

I. 79: Our analysis focuses exclusively on the 173,799 segments of the SWOT River Database³⁵
v16 that represent river reaches and excludes all segments associated with lakes and
reservoirs.

I. 431: We only compute river storage at SWORD reaches representing rivers (type 1 and type
5), removing reaches corresponding to lakes, reservoirs, dams/waterfalls, and ghost reaches³⁵.
**Altenau et al.³⁵ describes how reservoirs were inventoried in SWORD. In brief, the SWOT Prior**
**Lake Database (PLD), described in full by Wang et al. (2025), provides boundaries for**
**approximately 6 million lakes based on heavily curated Landsat data. This database was**
**intersected with the SWORD centerline and the resulting reaches were flagged as lakes. These**
**reaches were excluded from our analysis.**

4---A major weakness of this paper is the lack of a comprehensive quantification of uncertainty
in its estimates. Beyond the short study period (mentioned above), uncertainty arises from
several other sources:

(a) The reliability of the SWOT product at a global scale is not fully assessed: This is the first
global-scale scientific analysis using the SWOT L2 river product. Its performance in estimating
river water level and width, particularly its ability to capture seasonal variability, requires a
thorough global validation. The authors are encouraged to perform cross-validation using data
from traditional altimetry missions with longer time series (e.g., Jason, Sentinel series) for a
selection of representative reaches to assess WSE reliability. Furthermore, the accuracy of
SWOT-derived width could be validated against high-resolution Sentinel-1 SAR imagery. This
would also help quantify the potential error introduced by SWOT's effective 28-day revisit period
in capturing flood processes, especially flood peaks.

Thank you for your comment. We want to highlight two very important items regarding your
point:

- 1. As you suggested, we added a new full **uncertainty quantification of our river**
**storage anomaly (RSA) estimates** in this revised version of the manuscript (with
updated numbers in the text and the figures, and details in the Methods), from
uncertainties in WSE and width. For this, we **re-ran our full study with updated SWOT**
**uncertainty quantification** based on a previously published and verified methodology
(Coss et al., 2023) incorporated in FLaPE-Byrd. The global Δ RSA estimate is now $313 \pm$
129 km^3 for instance (uncertainties are also provided at the reach and basin scales).
This was one of the most time-consuming aspect of this revision but adds definite
robustness to our study.
- 2. The validation of SWOT's performance and requirements with other technologies and
measurements is an ongoing investigation undertaken by JPL/CNES and several groups
worldwide and represents a major undertaking. The responsibility of the SWOT "Science
Team" is to help advance SWOT's intended scientific goals, and we therefore analyze
the data and produce river water storage change estimates based on the current best
knowledge, including the latest uncertainties, regardless of their magnitude relative to
requirements. While **comprehensive validation papers are currently in review**, we
would be happy to share the results if it would be helpful. However, we cannot include
these extensive analysis in the present paper and assessing the validity of the stated
requirements is beyond the scope of this study.
**I. 190-191: While assessing compliance with SWOT mission requirements is**
**beyond our scope, we hope these results will contribute to advancing the**
**mission's scientific objectives.**

(b) The simplified processing for ice-covered rivers is a significant source of error: The
manuscript describes a simple "forward-filling" method to handle data gaps during the winter
ice-covered season. This approach likely leads to a significant underestimation of storage
changes in these regions, as sub-ice flow continues. Given the large number of rivers in the

Arctic and on broad mountainous regions/plateaus, the neglected storage variability in these
areas could have a non-negligible impact on both the global total and regional patterns.

Excellent point. We agree that sub-ice flow continues. We do not stop it by forward-filling. We
simply keep storage constant from the last observed RSA, which is in line with hydrographs
observed on Arctic rivers (see below from GRDC in situ monthly discharge data in $\text{m}^3 \cdot \text{s}^{-1}$). The
forward-filling methodology was thoroughly built: initially, we had performed a linear fill
everywhere and every time, and then realized that such was leading to notable overestimations
in the late winter/early spring and giving wrong seasonal patterns in Arctic rivers. Therefore, we
opted to keep a stable RSA when the ice flag is on (which doesn't mean that the river stops
flowing).

We do agree that these rivers are overall an important source of uncertainty, as referenced
multiple times in the text (see other responses involving Arctic rivers), such as:

I. 117: [...] and the absence of data over frozen water induces biases in elevation-width
relationships for Arctic rivers.

And even now directly in the abstract:

17 L. 22-23: While the Amazon's 2024 record drought, the observational challenges in the
18 Arctic, and SWOT's revisit frequency almost certainly contribute to the discrepancy, [...]

(c) The reliability of the peak storage month is not evaluated: A key finding presented in Figure 2
is the month in which river storage peaks. However, the robustness of this finding has not been
independently verified. It is suggested that the authors validate this result using other multi-
source remote sensing datasets (e.g., time series of river width or water level derived from
optical or microwave sensors) in representative regions to strengthen the credibility of this
conclusion.

Thank you for this comment. We agree that independent validation of the peak storage month is
highly valuable. However, we respectfully disagree that it was not evaluated. We already
performed a full correlation analysis of the RSA time series with both gauge data (Ext. Data Fig.
3) and the MeanDRS model simulation (Ext. Data Fig. 9). Furthermore, the lagged correlation
analysis with MeanDRS (Ext. Data Fig. 10) provides a quantitative assessment of potential
temporal offsets between datasets, confirming that the phase of the seasonal storage cycle is
well captured in most basins, with larger deviations (up to ~5 months) mainly in Arctic rivers.

However, to follow your suggestion, we completed the Extended Data Figure 6 with **two new**
**graphs (6a and 6b)**, now showing the monthly RSA regional contributions for both SWOT and
MeanDRS, providing another comparison between the two products seasonality.

**I. 217: Regional contribution to monthly offset between SWOT and modeled RSA can be**
**visualized in Extended Data Fig. 6. [...] In addition, the Lena and Yenisei basins account**
**for most of the discrepancies between SWOT and the model from November to March**
**(Extended Data Fig. 6).**

While a full cross-validation using additional multi-sensor datasets (e.g., optical or microwave
river width or water level products) would indeed be insightful, conducting such an analysis at
the global scale is beyond the scope of the present study and would require substantial
additional data processing and harmonization efforts. We believe that the current set of
validations already provides a robust assessment of the reliability of the inferred peak storage
13 month.

***Specific comments:***

Line 11-25: As specified in the paper title, one of the major contribution of this manuscript lies in
the mapping of river bank shapes. However, the findings in such dimension are not described in
Abstract at all.

As detailed above (major comment 1), we chose to focus on estimating the first river water
storage change to advance SWOT's scientific objectives, and to compare these estimates with
the only currently available model-based estimates. Riverbank morphology, while very
promising to analyze further regionally, serves primarily as a means to that end. We did mention
in the abstract (which needs to be short and concise):

**I. 19: Clear patterns of riverbed shape and storage variability expectedly emerge across major**
**basins.**

Line 81: The analysis uses data from 126,674 reaches. Could you clarify what percentage this
represents of the total number of river reaches (type 1 and 5) in the SWOT v16 database?
This would provide better context for the study's spatial coverage from the outset.

The data analyzed (126,674 reaches) represents 73% of all reaches of type 1 and 5 (171,824 in
the SWOT data downloaded), as written in the main text:

**I.82: The resulting filtered SWOT records are used to infer functional relationships between the**
**measured river widths and elevations³⁹ at 126,674 reaches, together representing 73% of the**
**world's widest river reaches (Extended Data Fig. 2a).**

We also added an **Extended Data Table 1** providing the number of unique reaches and
observations left out from the successive data quality filtering steps in the Methods part (see
page 29 below).

Line 88-92: The statement notes that the derived channel shapes are for the bathymetry
between the lowest and highest water levels observed. Given that the study period (2023-2024)
included a historic drought in the Amazon, which contributes over 50% of the total observed

storage variability, the "lowest observed" levels there may be anomalously low. **How might this**
**affect the derived bank shapes and their representativeness of a more typical year? What about**
**other basins similarly with hydroclimatic extremes during the study period?**

Excellent point. Indeed, we are only mapping the riverbank shapes observed by SWOT between
October 2023 and September 2024. As a result, we do not capture the full bank shape, and
there is no reason to assume that these shapes are representative of a typical year. We made
sure to clarify this in the main text:

I. 111: **However, the shapes only reflect what SWOT observed during 2023-2024.** Notably,
floodplains appear to be rarely captured in the elevation-width relationships (Fig. 1). Results
also reflect early-stage limitations in both our three-domain piecewise approach and SWOT's
current measurement capabilities and uncertainties.

On the other hand, observing periods of historically low flows provides valuable information on
riverbank geometry at low stage, which is rarely accessible. With additional years of data, the
bank profiles will progressively be completed with other river stages and further refined:

I. 226: A longer data record from SWOT's planned 3-year duration will help refine the elevation–
width approach and measurement uncertainties, **and allow the analysis of potentially more**
**representative hydrological years.**

Line 109: The acknowledgment that "floodplains appear to be rarely captured" is critical. Since
many global hydrological models explicitly include floodplain storage, could this methodological
limitation be a primary driver of the discrepancy between SWOT-observed storage and modeled
estimates, rather than solely model parameter errors? This point is understated in the
discussion part.

This is a very good point. We did not clearly mention in the main text that the model chosen for
comparison (referred to MeanDRS only in the Methods) does not include floodplain storage. We
added this important piece of information, to highlight that the two datasets are as comparable
as possible:

I.206-208: We leverage a translational dataset⁵¹ to identify modeled river reaches that directly
correspond to the SWOT observations, together with publicly available **river water storage**
**(without floodplain)** model estimates³, to provide context from prior knowledge (Fig. 4a).

Line 125-127: The manuscript mentions that missing data are infilled. The methods (Line 518)
state a "forward filling approach is adopted when water is 'likely fully ice covered'". This
assumes storage is static under ice. However, sub-ice flow persists, and storage can change
significantly during winter. This assumption will likely lead to an underestimation of storage
variability in Arctic and boreal rivers. Has the potential magnitude of this underestimation on the
global total been assessed? May this challenging issue for the Arctic rivers be validated with in-
situ materials?

See response above to your major comment 4 (b).

Line 155-156: The Amazon basin accounts for 172.7 km³ of the total 313.4 km³ annual river
storage variability (ARSA), which is approximately 55%. Given the severity of the 2023-2024
drought, this single basin's anomalous condition heavily influences the global total. It seems
problematic to compare this single, anomalous year for the globe's most dominant basin against

a 30-year model climatology to draw conclusions about model performance. **The paper should**
**more strongly emphasize that the global value is likely not representative of a climatological**
**average.**

We couldn't agree more, and we further emphasized your valid point in the abstract and main
text, with stronger tone:

6 L. 22-23: While the Amazon's 2024 record drought, **the observational challenges in the**
7 **Arctic, and SWOT's revisit frequency almost certainly** contribute to the discrepancy, the new
observations point to distinct knowledge limitations in surface water science.

9 L. 218: Exceptional drought conditions in the Amazon basin since 2023⁵² **very** likely impacted
the observed global Δ RSA (Extended Data Fig. 6) [...]

I. 222: SWOT-based variability for 2023-2024 **is likely** rationally lower than that of past
climatological averages.

I. 237-240: **SWOT-based river storage change estimates still face numerous uncertainties,**
**but the mission offers a new path forward: with more observations across annual cycles,**
**uncertainties may be untangled in hope** to build more physically grounded, globally
consistent models of river dynamics.

17 L.630: Regional hydroclimatic variability (e.g., El Niño periods) exerts a substantial influence on
comparisons between a single year of SWOT observations and long-term gauge-corrected
simulated means, as the study period captured by SWOT (2023-2024) may represent conditions
that are significantly different from climatological averages.

Line 165-171: The validation uses 61 in situ gauges. Extended Data Fig. 3 shows very low or
even negative correlations for several major basins (e.g., Rio Paraná, Colorado, Khatanga-
Kotuy). These poor correlations in non-ice-affected, major basins raise concerns about the
reliability of the RSA estimates in those regions. Could the authors provide potential
explanations for these specific discrepancies?

**Yes indeed. We completed the relevant sentence in the main text:**

I. 173: Seasonal validation confirms that SWOT RSA appears reliable across most of the globe,
except for parts of the Arctic, **southern South America and the Western U.S.** (Extended Data
Fig. 3).

**Potential explanations are also acknowledged later, pointing to the heavily managed and highly**
**complex nature of these basins:**

I. 575: Discrepancies in basins such as the Paraná or the Colorado could indicate limitations in
SWOT's current ability to consistently delineate water extent in complex and heavily managed
hydrological settings.

**The Kathanga-Kotuy is affected by seasonal ice cover. The Colorado has one of the most**
**extensive system of dams, reservoirs, and aqueducts on the planet and almost its entire flow is**
**diverted. Approximately 130 hydroelectric dams and reservoirs have been built on the Upper**
**Paraná River, significantly altering its natural flow. Thus, comparing SWOT RSA signal at the**
**basin level to one gauge on one of these rivers can be relatively tricky.**

Line 193-209 & 218-223: in the main text, **the comparison between SWOT (10-2023 to 09-2024)**
**and Collins et al. (1980-2009) monthly river storage anomalies was merely made at the global**
**scale.** The results may be misleading to audience.

We completed Extended Data Figure 6 with **two new graphs (6a and 6b)**, now showing the
monthly RSA regional contributions for both SWOT and MeanDRS. The basins primarily
accounting for the monthly discrepancies are the Amazon, the Lena and the Yenisei (especially
from Nov to March).

**I. 217: Regional contribution to monthly offset between SWOT and modeled RSA can be**
**visualized in Extended Data Fig. 6.**

**I. 220: In addition, the Lena and Yenisei basins account for most of the discrepancies**
**between SWOT and the model from November to March (Extended Data Fig. 6).**

Line 427: The analysis retains SWOT observations flagged as 'suspect' and 'degraded'. What is
the rationale for including these lower-quality data points, and how might their inclusion impact
the accuracy of the final storage estimates?

The *reach_q* flag is a summary indicator. We removed only the 'bad' reach points because they
are the only ones that must be ignored according to JPL/CNES. From the SWOT product
description document: "*Measurements that are marked as 'bad' may be nonsensical and should*
*be ignored.*"

The philosophy of SWOT data quality flagging is that if any flag of potential concern at all is
triggered for any node in a reach, then the data in the entire reach cannot be marked as 'good'.
Therefore, reaches that have one node (out of 50) that might have a small issue are marked as
(at least) 'suspect'. Thus, most 'suspect' data and many 'degraded' data are, in fact, very
accurate.

Out of the 8,143,747 observations for reaches of type 1 and 5 in our dataset, **254,242 are 'bad'**,
**5,353,558 are 'degraded'**, **2,533,777 are 'suspect'**, and **only 2170 are 'good'**. If we had also
removed the 'degraded' points, we would have been left with **152,317 unique reaches and**
**only 2,535,947 observations**. As you can see, the number of data points flagged as 'degraded'
and 'suspect' are very numerous, as JPL/CNES have been very conservative with the data
quality flags.

Overall, we wanted to perform further selection based on other attributes (cross-over calibration,
dark water fraction, fraction of nodes available etc.). Please refer to your next comment for
further detail. We completed the Methods:

**I. 433: We filter SWOT observations with a summary quality indicator (reach_q) of 'bad' (3% of**
**observations), retaining observations deemed 'good' (0.03%), 'suspect' (31%), and 'degraded'**
**(66%).**

Line 435-438: A filter is applied to remove reaches where the observed elevation range exceeds
20 m. This seems like an arbitrary threshold that could systematically exclude some of the
world's most dynamic river reaches, particularly in large, low-gradient tropical basins or dam-
regulated reaches. Please provide a stronger justification for this filter or conduct a sensitivity
analysis to show its impact.

We added an **Extended Data Table 1** providing the number of unique reaches (1st line below)
and observations (2nd line) left out from the successive data quality filtering steps in the Methods
part.

Reaches SWOT files	rch_type = 1 or 5	reach_q < 3	xovr_cal_q = 0	dark_fra c ≤ 0.3	ice_clim_f = 0	obs_frac_n ≥ 0.5	xtrk_dis t ∈ [10-60] km	Reaches with ≥ 5 obs	WSE range < 20 m
215,485	171,824	170,791	170,675	170,079	169,914	164,105	158,176	131,344	126,674
9,948,845	8,143,747	7,889,505	4,746,552	3,527,809	2,905,972	2,076,785	1,788,292	1,716,718	1,646,813

- In the last step, that you mention, we performed a sensitivity analysis:
- By removing WSE ranges above 25 m, we end up with 128,499 reaches (i.e., -2.2% from
- 131,344).
- By removing WSE ranges above 20 m, we end up with 126,674 reaches (i.e., -3.6%).
- By removing WSE ranges above 15 m, we end up with 123,420 reaches (i.e., -6.0%).
- By removing WSE ranges above 10 m, we end up with 116,243 reaches (i.e., -11.5%).
- By removing WSE ranges above 5 m, we end up with 95,984 reaches (i.e., -26.9%).

As justified in the methods, we chose 20 m as a threshold because the largest flood height
variations observed in the Amazon⁵⁸ are about 11-12 m, and can be up to 15 m. By choosing 25
10 m instead, we would have retained 1,800 additional reaches that have WSE variations between
11 20 and 25 meters, some of which might have been right, but most of which we would not trust to
12 have been processed accurately.

In any case, the result is that we did not process those few reaches (1.4% of reaches), and we
might miss a little bit of global RSA range. We believe this doesn't impact the scientific
soundness of our study.

Line 446-448: The effective sampling rate is one observation every 28 days after filtering. This is
significantly sparser than the nominal revisit time. As acknowledged in Lines 714-723, such a
low sampling frequency is likely to miss the peaks of many flood events, leading to a systematic
underestimation of storage variability. This limitation seems significant enough that it should be
highlighted more prominently in the main text as a major contributor to the lower-than-modeled
storage values.

You are right, this is a significant limitation. As such, we highlighted it further in the abstract and
at the end of each section of the main text:

25 L. 22-23: While the Amazon's 2024 record drought, **the observational challenges in the**
26 **Arctic, and SWOT's revisit frequency almost certainly** contribute to the discrepancy, [...]

27 L. 115: In addition, one year of observations at an average repeat of 28 days (after data filtering,
Extended Data Fig. 1) likely obscures occasional high extremes [...]

**I. 177-179: Overall, SWOT's limited effective temporal sampling can cause the monthly**
**and annual RSA ranges to miss part of the true variability.**

Line 472-482: The EIV regression uses an assumed WSE uncertainty of 0.1 m and width
uncertainty of 30 m based on mission requirements. Since the paper notes the product's error
estimates are unreliable, how sensitive are the results (specifically the cross-sectional area
anomaly δA) to these assumed uncertainty values?

Thank you for this comment. In addition to **re-running the global scope with uncertainty**
**quantification, we also re-ran our study in ten contrasted and representative basins under**

**three additional degraded input uncertainty scenarios:** i) width uncertainty of 300 m instead
of 30 m, ii) elevation uncertainty of 0.5 m instead of 0.1 m, and iii) elevation uncertainty of 0.5 m
and width uncertainty of 300 m. These additional scenarios reflect potential SWOT degraded
measurement conditions. Please refer to this analysis in the **new Supplementary Information**
**section 4 and new Fig. S4.** In essence, we find that the main RSA estimates are robust to input
uncertainties, whose effects solely manifest through uncertainty propagation.

Line 483-496: The three-part piecewise fitting is used to represent flow regimes, including out-
of-bank flow. However, the main text states that floodplains are rarely captured. Please clarify
this apparent contradiction. **If out-of-bank flow is not being observed, what is the physical**
**interpretation of the third segment of the hypsometric curve, and is there a risk it is fitting to**
**measurement noise rather than a distinct flow regime?**

Depending on the river flow conditions when SWOT passed in 2023–2024, out-of-bank flow
may not have been captured. The three-part regression is designed to represent the three
stages described in the manuscript, but if some stages are never observed (because water
levels remain either too low or too high), the three domains will simply reflect whatever
conditions were actually sampled. These domains (i.e., the breakpoints) are first defined using
an ordinary least squares estimator. The slopes and intercept of the regressions are then
determined using the EIV approach. In some cases, the fitted slopes for all three segments are
identical, indicating that SWOT only sampled in-bank flow with an approximately constant
riverbank slope. In such cases, the three domains may not have distinct physical interpretations.
We completed the methods:

23 L. 504: The subdomains **ideally** reflect distinct hypsometric relationships across three potential
flow regimes: within-bank flow, the transition to out of bank flow, and out of bank flow³⁹.

**Depending on what SWOT sampled, only within-bank flow may have been captured or**
**kept post-filtering, and the three subdomains will then represent different in-bank**
**shapes.**

Line 495-502: For the 3% of reaches where a rectangular channel was assumed, this
simplification would inherently suppress storage variability (since width does not change with
WSE). What is the estimated impact of this simplification on the total global ARSA?

We emphasize that this approach does not suppress storage variability, since WSE still
changes, and a median width is assumed at all WSE rather than adjusting widths according to
water level (i.e., lower widths at lower WSE and higher widths at higher WSE). Unfortunately, by
definition, we cannot quantify the impact of enforcing rectangular fits because the counterfactual
is incomputable (rectangular fits are applied when width increases excessively with height ($m_i >$
$10,000$) or decreases ($m_i < 0$)).

I. 508: In some circumstances, the initial errors-in-variable regression fails to converge or
produces implausible results (**m_i terms either greater than 10,000 or negative**), [...]

**Referee #4 (Remarks to the Author):**

I co-reviewed this manuscript with one of the reviewers who provided the listed reports.

**Response to Referee 4:**

Dear Reviewer,

Thank you very much for your review. Please refer to the response corresponding to the report

you co-authored.